# LEARNING BY SHAKING: COMPUTING POLICY GRADIENTS BY PHYSICAL FORWARD-PROPAGATION

## ABSTRACT

Model-free and model-based reinforcement learning are two ends of a spectrum. Learning a good policy without a dynamic model can be prohibitively expensive. Learning the dynamic model of a system can reduce the cost of learning the policy, but it can also introduce bias if it is not accurate. We propose a middle ground where instead of the transition model, the sensitivity of the trajectories with respect to the perturbation (shaking) of the parameters is learned. This allows us to predict the local behavior of the physical system around a set of nominal policies without knowing the actual model. We assay our method on a custom-built physical robot in extensive experiments and show the feasibility of the approach in practice. We investigate potential challenges when applying our method to physical systems and propose solutions to each of them.

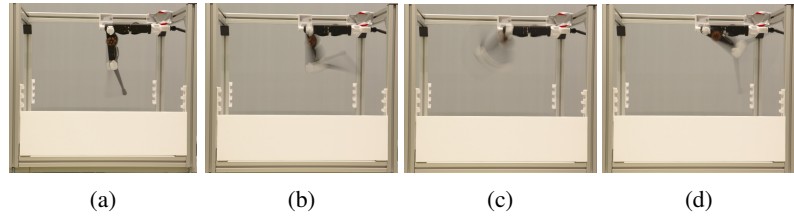

| (a) | (b) | (c) | (d) |

Figure 1: Physical finger platform in action with different policies.

## 1 INTRODUCTION

Traditional reinforcement learning crucially relies on *reward*(Sutton & Barto, 2018). However, reward binds the agent to a certain task for which the reward represents success. Aligned with the recent surge of interest in unsupervised methods in reinforcement learning (Baranes & Oudeyer, 2013; Bellemare et al., 2016; Gregor et al., 2016; Hausman et al., 2018; Houthooft et al., 2016) and previously proposed ideas (Schmidhuber, 1991a; 2010), we argue that there exist properties of a dynamical system which are not tied to any particular task, yet highly useful, and their knowledge can help solve other tasks more efficiently. This work focuses on the sensitivity of the produced trajectories of the system with respect to the policy so called *Physical Derivatives*. The term *physical* comes from the fact that it uses the physics of the system rather than any idealized model. We learn a map from the directions in which policy parameters change to the directions in which every state of the trajectory changes. In general, our algorithm learns the Jacobian matrix of the system at every time step through the trajectory. The training phase consists of physically calculating directional derivatives by the finite difference after applying perturbed versions of a nominal policy (a.k.a. controller). Perturbing the parameters of the controller is the reason for naming our method *shaking*. The test phase uses these directional derivatives to compute derivatives along unseen directions. Due to the difficulty of computing the Jacobian matrix by the finite difference in higher dimensions, we use random controllers joint with probabilistic learning methods to obtain a robust estimate of the Jacobian matrix at each instant of time along a trajectory. We are capable of this generalization to unseen perturbations because the trajectories of physical systems live on an intrinsic low-dimensional manifold and change slowly with the small changes in the parameters of the system (Koopman, 1931). This assumption holds as long as the system is not chaotic or close to a bifurcation condition (Khalil, 2002).

## 1.1 PRELIMINARIES

A reward function describes how close the agent is to the solution of the target task. In the absence of the reward, the agent will be given no means to find its way towards the solution. Let $\mathbf{x} \in \mathcal{X} \subseteq \mathbb{R}^d$ be a $d$-dimensional state vector that fully describes the environment with which the agent interacts. At each state, the agent is allowed to take action $\mathbf{u} \in \mathcal{U} \subseteq \mathbb{R}^q$ from a $q$-dimensional action space via a parameterised policy function $\mathbf{u} = \pi(\mathbf{x}; \boldsymbol{\theta})$. The agent will be rewarded $r(\mathbf{x}, \mathbf{u})$ by the function $r : \mathcal{X} \times \mathcal{U} \to \mathbb{R}$ when it takes action $\mathbf{u}$ at state $\mathbf{x}$. The goal of learning is to update $\boldsymbol{\theta}$ such that some desired target is achieved. The target can be anything as long as a concrete reward function is associated with it. In stochastic cases, *return* $R : \Pi(\Theta) \to \mathbb{R}$ is defined as a cumulative future discounted reward whose expectation is often of main interest. For parametric policies, the space of feasible parameters $\Theta$ has a one-to-one correspondence to the policy space $\Pi$. The agent who takes on the policy $\pi$ from state $\mathbf{x}_0$ produces the trajectory $\mathcal{T} \in \mathbb{T}$ where $\mathbb{T}$ is the space of possible trajectories. For a return function $R : \mathbb{T} \to \mathbb{R}$, the expected return becomes a function of the policy as $J(\pi_{\boldsymbol{\theta}}) = \mathbb{E}_{\mathcal{T}}\{R(\mathcal{T})\}$ where the expectation is taken with respect to the probability distribution $P(\mathcal{T}|\pi_{\boldsymbol{\theta}})$. There exist two major classes of approaches in reinforcement learning: value-based methods and value-free methods. In the first class, a surrogate function is defined to approximate the value of either a state $V(\mathbf{x})$ or a state-action pair $Q(\mathbf{x}, \mathbf{u})$. The policy is updated such that the agent tends towards states with higher values. The value-free methods update the policy directly without any need for an auxiliary function such as $V$ or $Q$. This paper mainly concerns the second class. The policy parameters are updated as

$$\boldsymbol{\theta}_{t+1} = \boldsymbol{\theta}_t + \alpha \left. \frac{\partial J(\pi_{\boldsymbol{\theta}})}{\partial \boldsymbol{\theta}} \right|_{\boldsymbol{\theta}=\boldsymbol{\theta}_t} \tag{1}$$

and the gradient $\partial J(\pi_{\boldsymbol{\theta}})/\partial \boldsymbol{\theta}$ is written as

$$\frac{\partial J(\pi_{\boldsymbol{\theta}})}{\partial \boldsymbol{\theta}} = \int_{\mathbb{T}} \frac{\partial p(\mathcal{T}|\pi_{\boldsymbol{\theta}})}{\partial \boldsymbol{\theta}} R(\mathcal{T}) \, \mathrm{d}\mathcal{T} \tag{2}$$

which is normally difficult to compute in practice. As can be seen in eq. (2), the integrand of the r.h.s. consists of two terms. The second term $R(\mathcal{T})$ is the return which is defined according to the target task. Hence, this term is task-dependent. The first term $\partial p(\mathcal{T}|\pi_{\boldsymbol{\theta}})/\partial \boldsymbol{\theta}$ though shows how the trajectories change with respect to a change in the policy. Notice that there is no notion of reward or any task-dependent quantities in this term. For an empirical distribution $p_e(\mathcal{T}|\pi) = \frac{1}{M} \sum_{i=1}^{M} \delta(\mathcal{T} - \mathcal{T}^{(i)})$, the dependence of partial derivative of the distribtion of $\mathcal{T}$ on the partial derivative of $\mathcal{T}$ can be explicitly derived as

$$\frac{\partial p_e(\mathcal{T}|\pi_{\boldsymbol{\theta}})}{\partial \boldsymbol{\theta}} = \frac{1}{M} \sum_{i=1}^{M} u_1(\mathcal{T} - \mathcal{T}^{(i)}) \frac{\partial \mathcal{T}}{\partial \boldsymbol{\theta}} \tag{3}$$

where $u_1$ is the unit doublet function (derivative of the Dirac delta function). This examplary distribution makes it clear that the change in the distribution of trajetories relates to the change of the trajectories themselves. As an unsupervised object, $\partial \mathcal{T}/\partial \boldsymbol{\theta}$ is of main interest in this paper.

## 1.2 PHYSICAL DERIVATIVE

In this paper, we investigate the feasibility of learning a less explored unsupervised quantity, the so called *Physical Derivative* which is computed directly from the physical system. In abstract terms, we perturb the policy and learn the effect of its perturbation on the resulting trajectory. The difference from traditional RL whose algorithms are based on eq. (1) is the absence of a specified reward function. Instead, we generate samples from $\partial p(\mathcal{T}|\pi_{\boldsymbol{\theta}})/\partial \boldsymbol{\theta}$ of eq. (2) that makes it possible to compute $\partial J(\pi_{\boldsymbol{\theta}})/\partial \boldsymbol{\theta}$ for an arbitrary return function $R$. If the exact model of the system is known, control theory has a full set of tools to intervene in the system with stability and performance guarantees. When the system is unknown, one could identify the system as a preliminary step followed by a normal control synthesis process from control theory (Ljung, 2001). Otherwise, the model and the policy can be learned together in a model-based RL (Sutton, 1996) or in some cases adaptive control (Sastry & Bodson, 2011). We argue that learning physical derivatives is a middle ground. It is not model-based in the sense that it does not assume knowing the exact model of the system. Rather, it knows how the trajectories of the system change as a result of perturbing the policy

parameters. This differential information of the system has applications in many downstream tasks. This work focuses on the concept and introduction of physical derivatives and direct applications would go significantly beyond the scope of this work. Few potential applications are discussed with more details in appendix C.

***Our contributions***— In summary, the key contributions of the current paper are as follows:

- A method to generate training pairs to learn the map from the policy perturbations to the resulting changes in the trajectories.

- Learning the above map as a probabilistic function and showing that it generalizes to unseen perturbations in the policy.

- Use the inverse of the above map to perturb the policy in the desired direction to achieve certain goals without conventional RL methods.

- Use a physical custom-built robotic platform to test the method and propose solutions to deal with the inherent issues of the physical system to ensure the practicality of the method (see fig. 1 for images of the platform and and appendix A for technical details).

- The supplementary materials for the paper, including code and the videos of the robot in action can be found in `https://sites.google.com/view/physicalderivatives/`

## 2 METHOD

In this section, we describe our pipeline to estimate the physical derivatives and our proposed solutions to the inevitable challenges that are likely to occur while working with a real physical robot. We are interested in $\partial \mathcal{T}/\partial \boldsymbol{\theta}$ which denotes how a small change in the parameters $\boldsymbol{\theta}$ of the controller results in a different trajectory produced by the system. We normally consider a finite period of time $[0, T]$ and the trajectory is an ordered list of states $\mathcal{T} = [\mathbf{x}_0, \mathbf{x}_1, \ldots, \mathbf{x}_T]$ where the subscript shows the time step. Therefore, having $\partial \mathcal{T}/\partial \boldsymbol{\theta}$ is equivalent with having $\partial \mathbf{x}_t/\partial \boldsymbol{\theta}$ for every $t \in \{1, \ldots, T\}$. Notice that the initial state $\mathbf{x}_0$ is chosen by us. Hence we can see it either as a constant or as a changeable parameter in $\boldsymbol{\theta}$. We kept it fixed in our experiments.

Assume $\mathbf{x}_t \in \mathbb{R}^d$ and $\boldsymbol{\theta} \in \mathbb{R}^m$. Hence, $\nabla_{\boldsymbol{\theta}} \mathbf{x}_t = \partial \mathbf{x}_t/\partial \boldsymbol{\theta} \in \mathbb{R}^{d \times m}$ where the $t^{\text{th}}$ row of this matrix is $\nabla_{\boldsymbol{\theta}} x_{it} = (\partial x_{it}/\partial \boldsymbol{\theta})^{\mathsf{T}} \in \mathbb{R}^m$ showing how the $i^{\text{th}}$ dimension of the state vector changes in response to a perturbation in $\boldsymbol{\theta}$. The directional derivative of $x_{it}$ in the direction $\delta \boldsymbol{\theta}$ is defined as

$$\nabla_{\boldsymbol{\theta}}^{\delta \boldsymbol{\theta}} x_{it} = \langle \nabla_{\boldsymbol{\theta}} x_{it}, \frac{\delta \boldsymbol{\theta}}{|\delta \boldsymbol{\theta}|} \rangle. \tag{4}$$

If (4) is available for $m$ linearly independent and orthonormal directions, $\{\delta \boldsymbol{\theta}^{(1)}, \delta \boldsymbol{\theta}^{(2)}, \ldots, \delta \boldsymbol{\theta}^{(m)}\}$, the directional derivative along an arbitrary $\delta \boldsymbol{\theta}$ can be approximated by

$$\nabla_{\boldsymbol{\theta}}^{\delta \boldsymbol{\theta}} x_{it} = \sum_{j=1}^m c_j \langle \nabla_{\boldsymbol{\theta}} x_{it}, \delta \boldsymbol{\theta}^{(j)} \rangle \tag{5}$$

where $c_j = \langle \delta \boldsymbol{\theta}, \delta \boldsymbol{\theta}^{(j)} \rangle$ is the coordinates of the desired direction in the coordinate system formed by the orthonormal bases.

In practice, $m$ directions $\delta \boldsymbol{\theta}^{(j)}$ can be randomly chosen or can be along some pre-defined axes of the coordinate system. To compute $\langle \nabla_{\boldsymbol{\theta}} x_{it}, \delta \boldsymbol{\theta}^{(j)} \rangle$, the nominal policy parameters $\boldsymbol{\theta}$ are perturbed by $\delta \boldsymbol{\theta}^{(j)}$ as $\boldsymbol{\theta}^{(j)} \leftarrow \boldsymbol{\theta} + \delta \boldsymbol{\theta}^{(j)}$ and the derivative is computed as

$$\langle \nabla_{\boldsymbol{\theta}} x_{it}, \delta \boldsymbol{\theta}^{(j)} \rangle = \lim_{h \to 0} \frac{x_{it}(\boldsymbol{\theta} + h \delta \boldsymbol{\theta}^{(j)}) - x_{it}(\boldsymbol{\theta})}{h}. \tag{6}$$

This quantity is often approximated by finite difference where $h$ takes a small nonzero value. By perturbing the parameters $\boldsymbol{\theta}$ along $m$ orthonormal directions $\delta \boldsymbol{\theta}^{(j)}$ and computing the approximate directional derivative by (6), $\nabla_{\boldsymbol{\theta}}^{\delta \boldsymbol{\theta}} x_{it}$ can be computed along every arbitrary direction $\delta \boldsymbol{\theta}$, meaning that, we can compute $\nabla_{\boldsymbol{\theta}} x_{it}$ by evaluating it along any direction which is the aim of this paper.

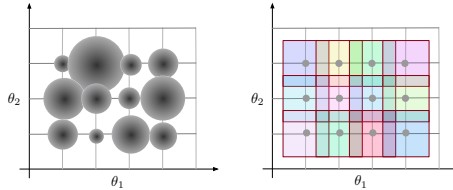

Figure 2: Gaussian (left) and uniform (right) shaking examples.

In the matrix form for $\mathbf{x} \in \mathbb{R}^d$, we can compute $\nabla_{\boldsymbol{\theta}}^{\delta\boldsymbol{\theta}^{(j)}}\mathbf{x} = [\nabla_{\boldsymbol{\theta}}^{\delta\boldsymbol{\theta}^{(j)}}x_1, \nabla_{\boldsymbol{\theta}}^{\delta\boldsymbol{\theta}^{(j)}}x_1, \ldots, \nabla_{\boldsymbol{\theta}}^{\delta\boldsymbol{\theta}^{(j)}}x_d]^\top$ in a single run by computing (6) for all $d$ dimensions of the states. Let's define

$$\Delta_{\boldsymbol{\theta}}\mathbf{x} \triangleq [\nabla_{\boldsymbol{\theta}}^{\delta\boldsymbol{\theta}^{(1)}}\mathbf{x}, \nabla_{\boldsymbol{\theta}}^{\delta\boldsymbol{\theta}^{(2)}}\mathbf{x}, \ldots, \nabla_{\boldsymbol{\theta}}^{\delta\boldsymbol{\theta}^{(m)}}\mathbf{x}] \quad (7)$$

where $\Delta_{\boldsymbol{\theta}}\mathbf{x} \in \mathbb{R}^{d \times m}$ and let $\Lambda = [\delta\boldsymbol{\theta}^{(1)}, \delta\boldsymbol{\theta}^{(2)}, \ldots, \delta\boldsymbol{\theta}^{(m)}]$. Therefore, if $\Delta_{\boldsymbol{\theta}}^{\delta\boldsymbol{\theta}}\mathbf{x}$ shows the directional derivative of $\mathbf{x}$ along $\delta\boldsymbol{\theta}$, we can write it as:

$$\nabla_{\boldsymbol{\theta}}^{\delta\boldsymbol{\theta}}\mathbf{x} = \Delta_{\boldsymbol{\theta}}\mathbf{x}(\Lambda^\top \delta\boldsymbol{\theta}) \quad (8)$$

which is only a vectoral representation of eq. (4). Even though the linear formula of eq. (8) requires only $m$ directional derivatives, it has two major downsides. First, it does not give a clear way to incorporate more than $m$ training directional physical derivatives. Second, the linear approximation remains valid only for very small $\delta\boldsymbol{\theta}$. We propose Gaussian Process (GP) as a nonlinear probabilistic function approximator (Rasmussen, 2003) to capture the maps $\hat{g}_t$ defined as

$$\hat{g}_t : \Theta \to \mathcal{X} \quad (9)$$
$$\hat{g}_t(\delta\boldsymbol{\theta}) = \delta\mathbf{x} \quad (10)$$

where subscript $t$ shows the function that maps $\delta\boldsymbol{\theta}$ to the change of the states $\delta\mathbf{x}_t$ at time step $t$. We considered distinct functions for every time step. Taking into account the commonality among the function approximators corresponding to different time steps is deferred to future research. Learning this map requires training data that comes from an initial data collection phase called *shaking*. Shaking refers to perturbing parameters of the controller to obtain the set of trajectories produced by the perturbed controllers.

The perturbation can be either regular or stochastic. Stochastic perturbations have the advantage over regular perturbations that the agent does not need to be worried about perturbing the parameters in a particular direction. Besides, in some cases, perturbing the parameters of the policy in certain directions is infeasible. We propose two methods of shaking called *Gaussian* and *Uniform* shaking.

*Gaussian shaking—* Likely values of $\boldsymbol{\theta}$ create nominal policies encoded by $\{\boldsymbol{\theta}^{(1)}, \boldsymbol{\theta}^{(2)}, \ldots, \boldsymbol{\theta}^{(m)}\}$. We put Gaussian distributions centered at each of the nominal values resulting in a mixture of Gaussians. To reduce the hyper-parameters, we assume the variances of the Gaussians are themselves sampled from an exponential distribution making sure they all take positive values (See fig. 2 left). Here, we manually choose a reasonable value for the rate parameter of the exponential distribution. Doing inference on the hyper-parameters of the sampling distributions can be a topic for future research especially in active learning for a more clever less costly sampling stratgey.

*Uniform shaking—* In this setting, the state space of the changeable parameters of the policy is discretized and a uniform distribution is assumed around each value of this grid with some overlapping with the neighboring cells (See fig. 2 right).

We show the effect of each of these sampling methods later in section 4. We observed that the results are less sensitive to the hyper-parameters of the uniform sampling than Gaussian sampling. A carelessly chosen rate for the exponential distribution that generates the variances of the Gaussians in Gaussian sampling can result in too local or global sampling that gives rise to a large variance or bias in the estimated gradients.

## 3 REAL WORLD CHALLENGES

In this section, we present two major low-level challenges that are common when dealing with physical systems. There exist inherent noise and imperfection in the system that results in a change in the produced trajectories while the policy parameters are kept fixed. In our finger platform, we observed two different major sources of noise which are likely to occur in other physical systems too. We call them *temporal* and *spatial* noise for the reasons that come in the following.

**Temporal noise.**  The temporal noise represented by $\mathbf{n}$ affects trajectories by shifting them in time

$$\mathbf{x}_t \leftarrow \mathbf{x}_t + \mathbf{n} \text{ for } t = 0, 1, \ldots, T. \tag{11}$$

Notice that the absence of subscript $t$ in $\mathbf{n}$ shows that this noise is not time-dependent, i.e., the time shift does not change along the trajectory as time proceeds.

**Spatial noise.**  The trajectories affected by spatial noise cannot be aligned with each other by shifting forward or backward in time. We can model this noise as a state-dependent influence on the state of the system at every time step.

$$\mathbf{x}_t \leftarrow \mathbf{x}_t + \mathbf{n}_{\mathbf{x}_t} \tag{12}$$

The following definition makes the distinction more concrete.

**Definition 1.** *Consider two trajectories $\mathcal{T}^{(1)}(t)$ and $\mathcal{T}^{(2)}(t)$ as two temporal signals. Assume $S_{t_\circ}$ is the shift-in-time operator defined as*

$$S_{t_\circ} \mathcal{T}(t) = \mathcal{T}(t + t_\circ) \tag{13}$$

*for an arbitrary function of time $\mathcal{T}(t)$. We say $\mathcal{T}^{(2)}(t)$ is temporally noisy version of $\mathcal{T}^{(1)}(t)$ if*

$$\exists t_\circ \in \mathbb{R} \ s.t. \ \|\mathcal{T}^{(2)} - S_{t_\circ} \mathcal{T}^{(1)}\|_1 \leq \epsilon \tag{14}$$

*where $\epsilon$ is a hyper-parameter threshold that reflects our prior confidence about the accuracy of the motors, joints, physical and electrical elements (in general construction process) of the robot. On the other hand, $\mathcal{T}^{(2)}$ is called a spatially noisy version of $\mathcal{T}^{(1)}$ if*

$$\nexists t_\circ \in \mathbb{R} \ s.t. \ \|\mathcal{T}^{(2)} - S_{t_\circ} \mathcal{T}^{(1)}\|_1 \leq \epsilon \tag{15}$$

### 3.0.1 SOLUTION TO TEMPORAL NOISE

Fortunately, this type of noise is not state-dependent by definition. If we find out how much a trajectory is shifted in time with respect to another trajectory, we can simply shift the trajectory for those many time steps and compensate for the delay. Hence, the problem becomes detecting the lagged trajectories with respect to a reference trajectory and also estimate the amount of the required time shift to compensate for the delay. We can either use physical landmarks in the trajectories to align them or use the correlation between them as a measure of alignment. The later gave better results, hence, we postpone the description of the former to the appendix D.1.

**Correlation-based delay estimation**  In this method, we use the correlation between zero-meaned trajectories $\mathcal{T}^{(i)}$ and $\mathcal{T}^{(j)}$ to check if one is the lagged version of the other one. The delay $\tau$ is found by

$$\tau^* = \underset{\tau}{\operatorname{argmax}} \sum_{t=0}^{T-\tau} \langle S_\tau \mathbf{x}_t^{(i)}, \mathbf{x}_t^{(j)} \rangle \tag{16}$$

where $S_\tau$ is a shift-operator by $\tau \in \mathbb{Z}$ time steps. In practice, we take one trajectory of $\{\mathcal{T}^{(1)}, \mathcal{T}^{(2)}, \ldots, \mathcal{T}^{(M)}\}$, e.g. $\mathcal{T}^{(r)}$ as the reference and synchronize other trajectories with respect to it using eq. (16). The trajectories must be initially normalized to avoid trivial solutions where every trajectory is pushed towards the larger parts of the reference trajectory. For illustrative purposes, the plots of fig. 14 show a sample of the lagged trajectory from the finger platform and its correction by the above method.

### 3.1 SOLUTION TO SPATIAL NOISE

The spatial noise can be a stochastic function of the actuator, environmental change, and electronic drivers. In a perfect model of the transition dynamics $\mathbf{x}_{t+1} = f(\mathbf{x}_t, \mathbf{u}_t)$, applying the same control sequence $\{\mathbf{u}_0, \mathbf{u}_1, \ldots, \mathbf{u}_{T-1}\}$ always results in the same sequence of states $\{\mathbf{x}_1, \mathbf{x}_2, \ldots, \mathbf{x}_T\}$ when it starts from the same initial state $\mathbf{x}_0$. This assumption is often violated in physical systems as different runs of the same policy may result in different trajectories as can be seen in fig. 10 in the Appendix. The noise in the dynamics can be any function of states, input, and time. Therefore, it is difficult to model this noise since it requires a prohibitively large number of random experiments. The good news is that if the physical system is built properly, the effect of this noise is expectedly low. Based on our observations from the finger platform, we can assume the following.

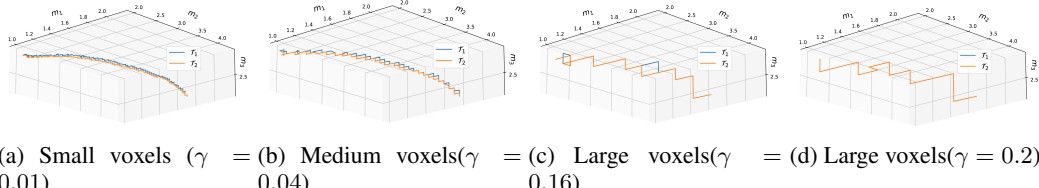

(a) Small voxels ($\gamma$ = 0.01)

(b) Medium voxels($\gamma$ = 0.04)

(c) Large voxels($\gamma$ = 0.16)

(d) Large voxels($\gamma = 0.2$)

Figure 3: The effect of voxels on supressing spatial noise of the physical system. The trajectories are produced by linear open-loop controllers as those in section 4.1 for the purpose of illustrating the effect of voxelization.

**Assumption 2.** *Limit on the physical noise: Let's the control sequence* $\mathbf{U} = \{\mathbf{u}_0, \mathbf{u}_1, \ldots, \mathbf{u}_{T-1}\}$ *be applied to the system $M$ times resulting in multiple sequence of states* $\mathcal{T}^{(1)}, \mathcal{T}^{(2)}, \ldots, \mathcal{T}^{(M)}$. *There exists a relatively small $\zeta$ such that*

$$\|\mathcal{T}^{(i)} - \mathcal{T}^{(j)}\|_\infty \leq \zeta \ \text{ for every } \ i, j \in \{1, 2, \ldots, m\}. \tag{17}$$

The word *relatively* here means that the change of the trajectory due to the inherent physical noise of the system must be small compared to the change of the trajectories when the parameters of the policy are perturbed.

To reduce the sensitivity of the estimated gradient to this unwanted spatial noise, we divide the state space of the physical system into regularly located adjacent cells called *voxels*. Each voxel $vox(\mathbf{c})$ is represented by its center $\mathbf{c}$ and is defined as

$$vox(\mathbf{c}) = \{\mathbf{x} \in \mathcal{X} \mid \|\mathbf{x} - \mathbf{c}\|_\infty \leq \gamma\} \tag{18}$$

where $\gamma$ is the parameter of the voxelization. The concept of the voxel is roughly used as a *superstate*. Every state that ends up within $vox(\mathbf{c})$ gives rise to the same superstate. After recording the trajectories form the robot, every state is mapped to the center of the voxel it belongs to as

$$\mathbf{c} \leftarrow \mathbf{x} \text{ for } \mathbf{x} \in vox(\mathbf{c}) \tag{19}$$

After voxelization, we work with $\mathbf{c}$ instead of $\mathbf{x}$. For example, all the gradients of (7) are computed as $\nabla_{\boldsymbol{\theta}} \mathbf{c}$ rather than $\nabla_{\boldsymbol{\theta}} \mathbf{x}$. To illustrate the positive effect of voxelization of the state space, it can be seen in fig. 3 that increasing the voxel size improves the overlapping between two trajectories that deviate from each other due to the inherent spatial noise of the system not because of perturbing the parameters of the policy, but because of the inherent imperfection of the mechanical and electrical components of the system. This benefit comes with a cost which is the error introduced by voxelization. Fortunately, this error is bounded due to the following lemma

**Lemma 3.** *The error caused by voxelization is bounded and inversely proportional to the size of each voxel (see appendix F.1 for a brief proof).*

After dealing with the challenge of inherent noise, we pursue the main goal of this paper which is estimating $\partial\mathcal{T}/\partial\boldsymbol{\theta}$ directly from the physical system. In the following, we investigate the use of the different type of controllers to emphasize the extent of applicability of the proposed method.

## 4 EXPERIMENTS

In this section, we show how physical derivatives can be estimated in practice through several experiments. Notice that our work is different from computing gradients around the working point of a system by finite-difference. We aim to collect samples from such gradients by perturbing a grid of nominal values of the policy parameters and then generalize to unseen perturbations by Gaussian process as a probabilistic regression method. The experiments are designed to show each challenge separately and the efficacy of our proposed solution to it. Due to space constraints, details to the physical platform can be found in section A in the Appendix. See[1] for videos of the robot while collecting data for different experiments and more backup materials.

---

[1] https://sites.google.com/view/physicalderivatives/

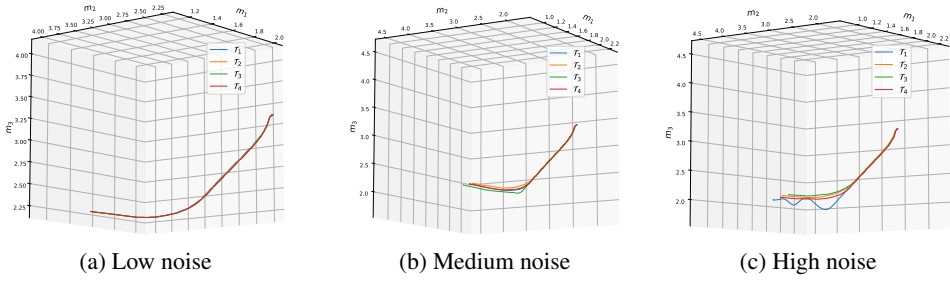

(a) Low noise         (b) Medium noise         (c) High noise

Figure 4: PD controller with various noise intensities on $K_p$ parameter.

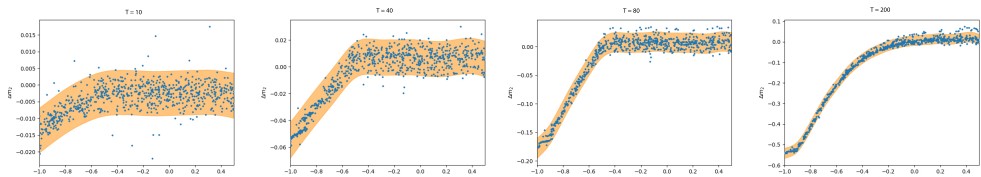

Figure 5: The time evolution of the GP approximated $\hat{g}_t$ for PD feedback controller at some exemplary time instances

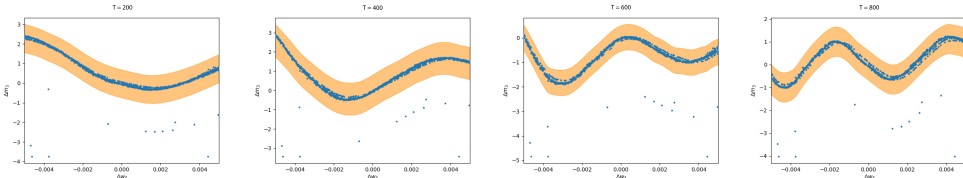

Figure 6: The time evolution of the GP approximated $\hat{g}_t$ for nonlinear sinusoidal open-loop controller at some exemplary time instances.

## 4.1 LINEAR OPEN-LOOP CONTROLLER

As a simple yet general policy, in this section, we consider an open-loop controller which is a linear function of time. The policy $\mathbf{u}_t = [u_{1t}, u_{2t}, u_{3t}]$ constitutes the applied torques to the three motors $\{m_1, m_2, m_3\}$ of the system and is assigned as

$$u_{it} = w_i t + b_i \quad \text{for} \quad i = 1, 2, 3 \tag{20}$$

Notice that the torque consists of two terms. The first term $w_i t$ grows with time and the second term remains constant. The controller has 6 parameters in total denoted by $\boldsymbol{\theta}$. The task is to predict $\nabla_{\boldsymbol{\theta}} \mathbf{x}_t$ for every $t$ along the trajectory. In the training phase, the training data is obtained via shaking as described in section 2.

fig. 7 shows examples of nominal trajectories + trajectories produced by the perturbed controller and the computed derivatives. The arrows are plotted as they originate from the perturbed trajectories only for easier distinction. Each arrow corresponds to the change of the states at a certain time step on the source trajectory as a result of perturbing the policy. Each figure corresponds to a pair of nominal values of $\{w, b\}$ for the linear open-loop controller. See fig. 29 for examples.

## 4.2 NONLINEAR OPEN-LOOP CONTROLLER

Physical derivatives can naturally be computed for either linear or nonlinear controllers which makes it different from taking the gradient of models through time. In model-based methods, if the model's transition dynamics is not differentiable, taking the derivative is theoretically challenging. However, our method is taking advantage of the real physics of the system to compute the gradients regardless of whether the approximating model is differentiable or not. To elaborate more on this, we test our method for a simple but nonlinear policy, i.e., $u_t = \mathcal{A} \sin(\omega t)$. The sinusoidal torque is applied

Table 1: The aggregate performance of our method to predict physical derivatives in unseen directions of perturbations to the parameters. $\langle \cdot \rangle$ shows time average. The first column is the normalized time averaged MSE. The second column is the time averaged GP score (closer to 1 is better. See appendix E.4 for definition). The third column is the time averaged misalignment between derivatives. Every experiment is repeated for 10 voxel sizes and the values are chosen for the best voxel size. N: Gaussian sampling, U: uniform sampling.

| Task | $\langle \text{MSE} \rangle$ | $\langle \text{Score} \rangle$ | $\langle \cos \alpha \rangle$ |
|---|---|---|---|
| PD controller (N) | 0.00871 | 0.8991 | 0.9492 |
| PD controller (U) | 0.0018 | 0.9841 | 0.9723 |
| Sine 2 joints (U) | 0.0368 | 0.7992 | 0.9513 |
| Sine 2 joints (N) | 0.0796 | 0,6696 | 0,9553 |

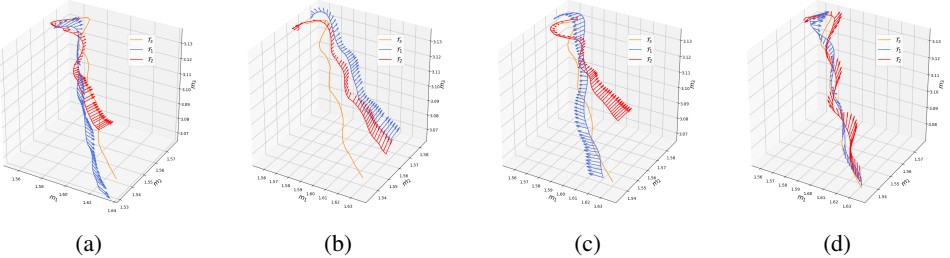

(a)      (b)      (c)      (d)

Figure 7: Pysical gradients computed for various time steps along a source trajetcory using two perturbed trajectories of linear open-loop controller.

to either one or two motors of the system to investigate the performance of our method. We tested Gaussian and uniform shaking for $\theta = \{\mathcal{A}, \omega\}$ as parameters of this controller. The GP interpolation for the partial derivatives at some time instances along the trajectory can be seen in fig. 6 and more extensively in figs. 16 to 18 in the Appendix. One might be interested in the direction of the predicted derivative instead of its exact size. To this end, we take several test perturbations for every time step and use $\cos(\alpha)$ as a measure of alignment between the predicted and ground-truth derivative vectors. The time evolution of the histogram of this measure along the trajectory shows a better alignment as time proceeds. This effect can be seen in figs. 27 and 28. This confirms our observation of initial transient noise in the system that dies out gradually by the progression of time. The overall performance of our method in predicting physical derivatices in unseen directions for two different shaking methods is shown in appendix E.

### 4.3 FEEDBACK CONTROLLER

Often in practice, the policy incorporates some function of the states of the system. Some well-known examples which have been extensively used in control applications are P, PD, PI and PID controllers. Here, we consider two member of this family, i.e., P and PD controllers. The policy becomes $\mathbf{u} = K_p \mathbf{e}$ for P controllers and $\mathbf{u} = K_p \mathbf{e} + K_d \dot{\mathbf{e}}$ for PD controllers. The error $\mathbf{e}$ shows the difference between the current state $\mathbf{x}$ and the desired state $\mathbf{x}^*$. The parameters of the controller $\{K_p, K_d\}$ are scalar values that are multiplied by the error vector element wise. This implies that the controller parameters are the same for three motors leaving the controller of the whole platform with two parameters that weights the value and the rate of the error. We applied the uniform and Gaussian shaking for the set of parameters $\theta = \{K_p, K_d\}$ with different scenarios. The GP interpolation for the physical derivatives at some time instances along the trajectory can be seen in fig. 6 and more extensively in figs. 19 to 24 in the Appendix. The time evolution of the histogram of misalignment between predicted and ground-truth directional derivatives (see figs. 25 and 28 in the appendix) once again confirms the existence of the initial transient noise as was also observed in the section 4.2. Similar to the sinusoidal experiment, the overall performance of our method is presented in appendix E.

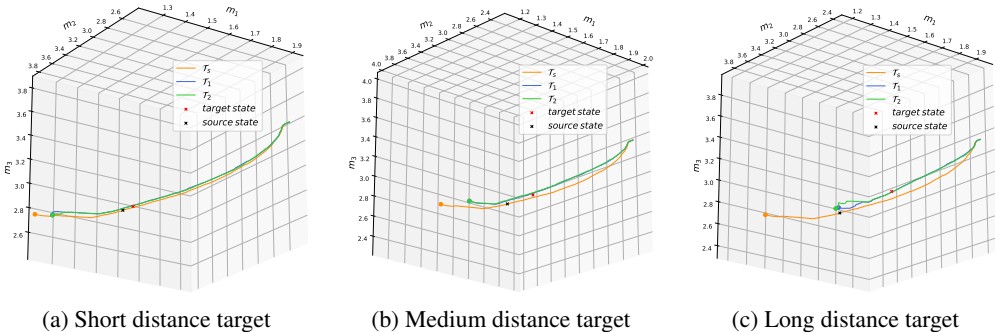

(a) Short distance target          (b) Medium distance target          (c) Long distance target

Figure 8: Zero-shot planning with constraint satisfaction. The orange trajectory is the source produced by the nominal controller. The green and blue are two sampled trajectoreis that are produced by perturbing $k_p$ to $k_p^*$ by eq. (21)

### 4.4    ZERO-SHOT PLANNING TASK

Our previous experiments in sections 4.1, 4.2 and 4.3 showed that learning the physical derivative map is feasible for various types of controllers. In this section, we demonstrate an example of a constrain satisfaction task by means of the physical derivative map. In this experiment, the superscript $(s)$ corresponds to the nominal trajectory which is called *source*. Assuem the system is controlled by a PD controller to reach a target state $\mathbf{x}^*$, i.e., the control torques are designed as $\mathbf{u} = k_p^{(s)}(\mathbf{x} - \mathbf{x}^*) + k_d^{(s)}\dot{\mathbf{x}}$. The controller does a decent job to reach the target state given reasonable values for $k_p$ and $k_d$. However, such controller does not give us a clear way to shape the trajectory that starts from $\mathbf{x}_\circ$ and ends at $\mathbf{x}^*$. Assume it is desired that the nominal controlled trajectory $\mathcal{T}^{(s)}$ passes through an intermediate state $\mathbf{x}_t^*$ at time $t$ on its way towards the target state $\mathbf{x}^*$ (we can equally assume that the system must avoid some regions of the state space because of safety reasons). The solution with physical derivatives is as follows . Assume $k_d^{(s)}$ is fixed and only $k_p^{(s)}$ is changeable. If the physical derivatives map is available, we have access to $\hat{g}_t(k_p^* - k_p^{(s)}) = (\mathbf{x}_t^* - \mathbf{x}_t^{(s)})/(k_p^* - k_p^{(s)})$. By simple algebraic rearrangement, we have

$$k_p^* = \frac{\mathbf{x}^* - \mathbf{x}_t^{(s)}}{\hat{g}_t(k_p^* - k_p^{(s)})} + k_p^{(s)}. \tag{21}$$

The new parameter of the policy is supposed to push the source trajectory $\mathcal{T}^{(s)}$ towards a target trajectory $\mathcal{T}^*$ that passes through the desired state $\mathbf{x}_t^*$ at time $t$. The result of this experiment on our physical finger platform can be seen in fig. 8.

### 4.5    RELATED WORKS

A truly intelligent agent must develop some sort of *general competence* that allows it to combine primitive skills to master a range of tasks not only a single task associated with a specified reward function. The major part of such competence come from unsupervised experiences. Animals use a similar competence to quickly adapt to new environments (Weng et al., 2001). and function efficiently soon after birth before being exposed to massive supervised experience (Zador, 2019). Due to its generality, such basic skills can be inherited over generations rather than being learned from scratch (Gaier & Ha, 2019). Despite traditional RL that the learning is driven by an *extrinsic* reward signal, *intrinsically* motivated RL concerns task-agnostic learning. Similar to animals' babies (Touwen et al., 1992), the agent may undergo a developmental period in which it acquires reusable modular skills (Kaplan & Oudeyer, 2003; Weng et al., 2001) such as curiosity and confidence (Schmidhuber, 1991a; Kompella et al., 2017). Another aspect of such general competence is the ability of the agent to remain *safe* during its learning and deployment period (García & Fernández, 2015). In physical systems especially continuous control, *stability* is a major aspect of safety that implies states of the system converge to some invariant sets or remain within a certain bound (Lyapunov, 1992). Control theory often assumes the model of the system known to guarantee stability (Khalil, 2002). In the absence of the model, model-based RL learns the model along with

the policy. Hence, learning the transition model to predict the states in the future can be another intrinsic reward.

From a technical point of view, our work is relevant to sensitivity analysis and how it is used to train the parameters of models such as in Chen et al.'s NeuralODE. The method seemed to be effective in many tasks including learning dynamics (Rudy et al., 2019) , optimal control (Han et al., 2018), and generative models (Grathwohl et al., 2018). Our method can be seen as a mode-free sensitivity analysis in real-world systems. In NeuralODE, the gradient with respect to the parameters requires solving ODEs for both states and adjoint states that require a transition model. Since we are working directly on the physical system, we don't need to calculate the integrals forward in time. The systems itself acts as a physical ODE solver. We refer to appendix F for a more detailed review of the related works.

## 5 CONCLUSION

In this paper, we present a method to learn the way that the trajectories of a physical real world dynamical system changes with respect to a change in the policy parameters. We tested our method on a custom-built platform called finger robot that allows testing a couple of controllers with various settings to show the applicability of our method for linear, nonlinear, open-loop, and feedback controllers. By estimating the physical derivative function, we showed that our method is able to push a controlled trajectory towards a target intermediate state. We investigate the real-world challenges when doing a fine sensitive task such as estimating physical derivatives on a real robot and proposed solutions to make our algorithm robust to inherent imperfection and noise in physical systems. We focused mainly on low-level issues of physical derivative and showing the feasibility of estimating it robustly. We expect that physical derivatives will contribute to research areas such as safety, control with constrain satisfaction and trajectory planning, robust or safe control.

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

## A PHYSICAL PLATFORM

In this section, we introduce the physical robot on which we tested our method. The robot is called *finger platform* or simply *finger* throughout this paper. The range of movement for the motors are $[0, \pi]$, $[0, \pi]$, $[0, 2\pi]$ respectively. The axes of the plots throughout the paper are in radian. It consists of three articulated arms with three degrees of freedom in total (see fig. 9d). The motors $\{m_1, m_2, m_3\}$ are depicted in the figure. This naming remains consistent throughout this paper. Each arm is moved by a separate brushless DC motor and has one degree of freedom to swing in its own plane (see fig. 9a). Each arm is equipped with an encoder that measures its angle (see fig. 9b). The brushless motors are controlled by an electronic driver that receives torque values applied to each motor from a computer terminal via a CAN bus and applies the torques to the motors(see fig. 9c). Due to the imperfections of the arms, motors, and drivers, we did not use any model for the system including the inertial matrix of the robot or the current-torque characteristic function of the motors. The low-cost and safe nature of this robot makes it a suitable platform to test the idea of physical derivatives that requires applying many different controllers in the training phase.

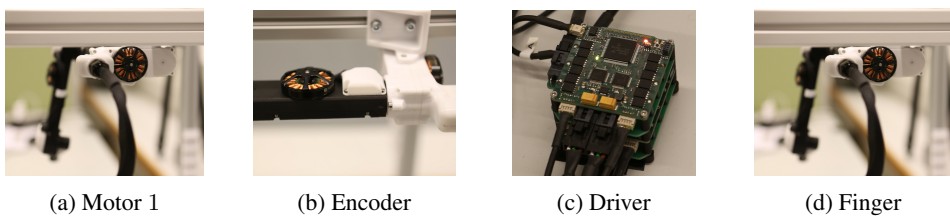

(a) Motor 1      (b) Encoder      (c) Driver      (d) Finger

Figure 9: Components of the physical finger platform

## B ADDITIONAL PLOTS ILLUSTRATING REAL WORLD CHALLENGES (SECTION 3)



(a) $t = 200$      (b) $t = 400$      (c) $t = 600$      (d) $t = 800$      (e) $t = 1000$

Figure 10: Same controller applied for multiple runs. The trajectories are produced by the linear open-loop controller similar to those used in section 4.1. See the plot for a different set of nominal parameters of the controller in fig. 13 in the Appendix (Zooming is recommended)



(a) $t = 200$      (b) $t = 400$      (c) $t = 600$      (d) $t = 800$      (e) $t = 1000$

Figure 11: The same as fig. 10 but for a different setting.



(a) $t = 200$     (b) $t = 400$     (c) $t = 600$     (d) $t = 800$     (e) $t = 1000$

Figure 12: Noisy linear open-loop controller



(a) $t = 200$     (b) $t = 400$     (c) $t = 600$     (d) $t = 800$     (e) $t = 1000$

Figure 13: The same as fig. 12 but for a different nominal parameters of the policy.

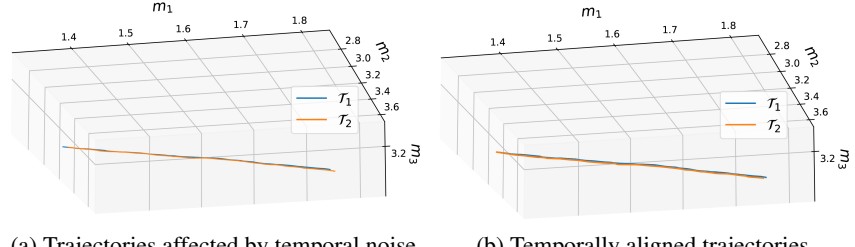

(a) Trajectories affected by temporal noise     (b) Temporally aligned trajectories

Figure 14: The effect of temporal noise in delaying one trajectory versus the other one and its correction. The trajectories are produced by the linear open-loop controller similar to those used in section 4.1(Zooming is recommended)

## C  APPLICATIONS OF PHYSICAL DERIVATIVES

If we know how the states of a trajectory change as a result of a change in the policy parameters, the policy can be easily updated to push the trajectory towards a desired one. For example, assume we are interested in going from the current trajectory $\mathcal{T}(\boldsymbol{\theta})$ to the target trajectory $\mathcal{T}^*$. The distance between these trajectories can get minimized by perturbing the policy parameters in the direction $-\partial\|\mathcal{T}(\boldsymbol{\theta}) - \mathcal{T}^*\|/\partial\boldsymbol{\theta}$. This direction is already available since we have estimated $\partial\mathcal{T}(\boldsymbol{\theta})/\partial\boldsymbol{\theta}$ as a physical derivative. As an exemplary case, we show this application of our method in practice in section 4. Other applications of physical derivatives are in *robust control* and *safety*. In both cases, the physical derivative allows us to predict the behaviour of the system if the policy changes in a neighbourhood around a nominal policy. Then, it is possible to make sure that some performance or safety criteria will not be violated for the local perturbation in the policy. As a concrete example, for an autonomous driving system, there can be a calibration phase during which physical derivatives of the car is estimated by perturbing the controller parameters around different nominal policies which are likely to occur in real roads. The calibration must be done in a safe condition and before deploying the system. When deployed, the estimated physical derivatives can be used to predict the effect of a change of the policy on the behaviour of the system and neutralize the change if it would move the car towards unsafe regions of its state space. The command that changes the policy can be issued by a high-level controller (e.g. guidance system), and the safety is confirmed by a low-level mechanisms through physical derivatives. This work focuses on the concept and introduction of physical derivatives and direct applications would go significantly beyond the scope of this work. In the following more detailed description of the use of physical derivatives in robust and safe control.

**Robust control**  In control theory, robust control relates to the design of a controller whose performance is guaranteed for a range of systems and controllers belonging to a certain neighborhood around the nominal system (Zhou & Doyle, 1998). It is desired to have a controller that keeps the performance of the system at a certain good level even if the parameters of the controller are not fixed to the theoretical values. Assume the performance of the system is associated with some function of a trajectory $\mathcal{E}(\mathcal{T})$. Changing the parameters of the controller $\boldsymbol{\theta}$ results in a change in the trajectories. This allows us to compute $\partial\mathcal{T}/\partial\boldsymbol{\theta}$ that consequently gives us $\partial\mathcal{E}(\mathcal{T})/\partial\boldsymbol{\theta}$ by the chain rule. Roughly speaking, between two sets of parameters $\boldsymbol{\theta}_1$ and $\boldsymbol{\theta}_2$, the set of parameters that gives the least $\partial\mathcal{E}/\partial\boldsymbol{\theta}$ is preferred. This means that by shaking the parameters of the controller and assessing the performance of the system, an estimate of the curvature of the landscape of $\mathcal{E}(\mathcal{T}(\boldsymbol{\theta}))$ is obtained. We prefer flatter regions of this space where a small change in $\boldsymbol{\theta}$ does not cause a drastic change in the performance metric $\mathcal{E}$.

**Safety**  Safety refers to the situations in which the agent may hurt itself or the environment and causes irreversible damages if it freely takes arbitrary actions (Garcıa & Fernández, 2015). For a safety-critical system whose full physical models are hard to obtain, the physical gradients can assist in avoiding restricting the parameters of the robot to avoid unsafe behavior. The physical derivatives are learned in the Lab environment before the robot is deployed into the wild. For example, a rover whose mission is to safely explore an unknown environment often enjoys a learning loop that allows it to adapt to the new environment. Even though the learning in the new environment requires sufficient exploration, the physical derivatives can be used to give a rough simulation of the robots next few states under a given update to its parameters. The potential harmful updates might be detected by such simulation and be avoided.

## D  EXTENDED SET OF SOLUTIONS TO THE REAL WORLD CHALLENGES

### D.1  DETECTING ZERO CROSSING

In this method, we take advantage of special landmarks in the trajectories. The landmarks are typically caused by physical constraints of the system. For example, when a robot's leg touches the ground, the velocity of the leg becomes zero. Likewise, when a joint reaches its physical limit, the velocity of the connected arm to the joint becomes zero or changes sign. In both cases, a zero crossing occurs that can be used as a landmark to synchronize lagged trajectories with a reference trajectory. Even though this method will eliminate the temporal noise, it requires the presence of such landmarks along the trajectories. Notice that from a mathematical point of view, there is nothing special about *zero*. We can pick any value of states along a reference trajectory and synchronize all other trajectories with respect to it. However, in practice, physical landmarks are easier to detect and have less ambiguity that consequently gives a more accurate synchronization.

## E  EXPERIMENTAL DETAILS

Starting position in all the experiments is $(\frac{\pi}{2}, \frac{\pi}{2}, \pi)$. Task's overall details are as following:

| Task | number of trajectories | timesteps |
|---|---|---|
| Linear (N) | 640 | 1500 |
| PD controller(N) | 640 | 1500 |
| PD controller(U) | 1000 | 1500 |
| Sine 1 joint(N) | 640 | 5000 |
| Sine 1 joint(U) | 1000 | 5000 |
| Sine 2 joints(U) | 640 | 5000 |
| Sine 2 joints(N) | 1000 | 5000 |

In normal sampling cases, we ran 10 simulations for each set of $\lambda$ parameters which indicates noise level.

### E.1 LINEAR

$$u_{it} = w_i t + b_i \quad \text{for} \quad i = 1, 2, 3 \tag{22}$$

#### E.1.1 GAUSSIAN SAMPLING

$$w_i = W_i + \epsilon_{w,i} \quad \text{for} \quad i = 1, 2, 3$$
$$\epsilon_{w,i} \sim N(0, e_w \times \|W_i\|_2)$$
$$e_w \sim exp(\lambda_w) \quad \text{for} \quad \lambda_w = 1, 5, 10, 50, 100, 500, 1000, 5000$$

$$b_i = B_i + \epsilon_{b,i} \quad \text{for} \quad i = 1, 2, 3$$
$$\epsilon_{b,i} \sim N(0, e_b \times \|B_i\|_2)$$
$$e_b \sim exp(\lambda_b) \quad \text{for} \quad \lambda_b = 1, 5, 10, 50, 100, 500, 1000, 5000$$

$$W = [0.00001, 0.0001, -0.00001], B = [-0.28, -0.15, -0.08]$$

### E.2 PD CONTROLLER

Final destination is $\left(\frac{\pi}{10}, 3\frac{\pi}{4}, 7\frac{\pi}{12}\right)$

#### E.2.1 GAUSSIAN SAMPLING

$$kp = KP + \epsilon$$
$$\epsilon_{kp} \sim N(0, e_{kp} \times \|KP\|)$$
$$e_{kp} \sim exp(\lambda_{kp}) \quad \text{for} \quad \lambda_{kp} = 1, 5, 10, 50, 100, 500, 1000, 5000$$
$$kd = KD + \epsilon$$
$$\epsilon_{kd} \sim N(0, e_{kd} \times \|KD\|)$$
$$e_{kd} \sim exp(\lambda_{kd}) \quad \text{for} \quad \lambda_{kd} = 1, 5, 10, 50, 100, 500, 1000, 5000$$

#### E.2.2 UNIFORM SAMPLING

$$kp \sim U(-0.5, 1.5), KP = 1$$

$$kd = KD = 0.01$$

### E.3 SINE 1 JOINT

#### E.3.1 GAUSSIAN SAMPLING

$$w = W + \epsilon$$
$$\epsilon_w \sim N(0, e_w \times \|W\|)$$
$$e_w \sim exp(\lambda_w) \quad \text{for} \quad \lambda_w = 1, 5, 10, 50, 100, 500, 1000, 5000$$

$$a = A + \epsilon$$
$$\epsilon_a \sim N(0, e_a \times \|A\|)$$
$$e_a \sim exp(\lambda_a) \quad \text{for} \quad \lambda_a = 1, 5, 10, 50, 100, 500, 1000, 5000$$

$$W = 0.01, B = 0.5$$

#### E.3.2 UNIFORM SAMPLING

$$w \sim U(0.005, 0.015), a = A = 0.5$$

### E.3.3 SINE 2 JOINTS

### E.3.4 GAUSSIAN SAMPLING

$$w_i = W_i + \epsilon \quad \text{for} \quad i = 1, 2$$

$$\epsilon_{w,i} \sim N(0, e_w \times \|W\|_2)$$

$$e_w \sim exp(\lambda_w) \quad \text{for} \quad \lambda_w = 1, 5, 10, 50, 100, 500, 1000, 5000$$

$$a_i = A_i + \epsilon \quad \text{for} \quad i = 1, 2$$

$$\epsilon_{a,i} \sim N(0, e_a \times \|A\|_2)$$

$$e_a \sim exp(\lambda_a) \quad \text{for} \quad \lambda_a = 1, 5, 10, 50, 100, 500, 1000, 5000$$

$$W = [0.01, 0.01], A = [-0.4, 0.5]$$

### E.3.5 UNIFORM SAMPLING

$$w_i \sim U(0.005, 0.015) \quad \text{for} \quad i = 1, 2, a = A = 0.5$$

### E.4 GP SCORE:

Definition of the GP score: The score is defined as $(1 - u/v)$, where u is the residual sum of squares $\Sigma(y_{\text{true}} - y_{\text{pred}})^2$ and $v$ is the total sum of squares $\Sigma(y_{\text{true}} - \text{mean}(y_{\text{true}}))^2$. The best possible score is 1.0.

### E.5 ZERO-SHOT PLANNING TASK:

For the task of section 4.4: Number of training trajectories: 100 each with 1500 time steps

Kd = 0.01

Kp = Uniformly sampled from $[0.2, 0.6]$

Initial point: $X_\circ = [\pi/2, \pi/2, \pi])$

desired position = $[\pi/10, 3 * \pi/4, 7 * \pi/12]$

## F DETAILED LITERATURE REVIEW

There has been a recent surge of interest in unsupervised methods in reinforcement learning when a task-specific reward function is not the only driving force to train the agent (Baranes & Oudeyer, 2013; Bellemare et al., 2016; Gregor et al., 2016; Hausman et al., 2018; Houthooft et al., 2016). A truly intelligent agent must behave intelligently in a range of tasks not only in a single task associated with its reward function. This requires the agent to develop some sort of *general competence* that allows it to come up with solutions to new problems by combining some low-level primitive skills. This general competence is a key factor in animals to quickly and efficiently adapt to a new problem (Weng et al., 2001). By calling the traditional RL, *extrinsicially motivated RL*, the new framework is called *intrinsically motivated RL*. There have been many ideas in this line with various definitions for the terms *motivation* and *intrinsic*. Some researchers assume a developmental period in which the agent acquires some reusable modular skills that can be easily combined to tackle more sophisticated tasks (Kaplan & Oudeyer, 2003; Weng et al., 2001). Curiosity and confidence are other unsupervised factors that can be used to drive the agent towards unexplored spaces to achieve new skills (Schmidhuber, 1991b; Kompella et al., 2017). Interestingly, there are observations in neuroscience that dopamine, a known substance that controls one's motivation for extrinsic rewards, is also associated with intrinsic properties of the agent such as novelty and curiosity. A novel sensory stimulus activates the dopamine cells the same way they are activated by extrinsic reward. Children build a collection of skills accumulatively while they engage in activities without a specific goal, e.g., hitting a ball repeatedly without a long-term target such as scoring a goal. The achieved skills contribute to their stability while handling objects (Touwen et al., 1992).

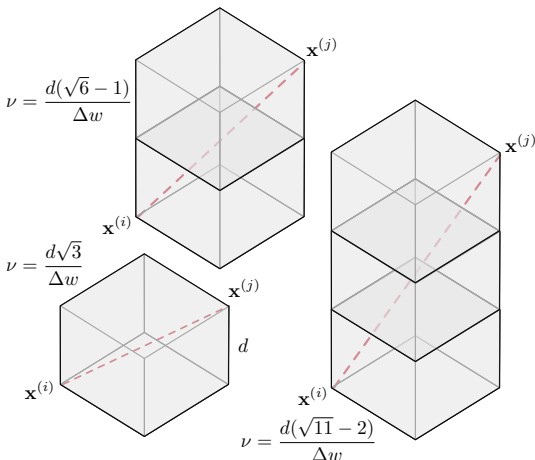

Figure 15: The maximum potential error in the estimated gradients when the space is voxelized. As can be seen, the error is vanishing when the corresponding voxels to $\mathbf{x}^{(i)}$ and $\mathbf{x}^{(j)}$ are far from each other.

Another line of work concerns the fundamental constraints of the agent/environment and ensures those constraints are met while learning. For example, in many practical systems, learning episodes must halt if the system is likely to undergo an irreversible change. For example, the training episodes of a fragile robot must ensure the robot does not fall or will not be broken in any circumstance while acting under a certain policy. The general name *safe RL* embodies ideas to tackle such issues in current interactive learning algorithms (Garcıa & Fernández, 2015). One major aspect of safety is *stability* that loosely means that states of the system converge to some invariant sets or remain within a certain bound (Lyapunov, 1992). Control theory enjoys a physical model of the system to guarantee stability (Khalil, 2002). When the physical model is not known in advance, the model is either learned along with the policy (model-based RL) or will be implicitly distilled in the value function (model-free RL) (Sutton & Barto, 2018). Stability can be categorized as an intrinsic motivation for the agent. No matter what task the agent aims to solve, it must remain stable all the time. Learning the transition model which is the major concern of model-based RL can also be seen as intrinsic motivation. The agent learns to predict the future step given the current state. The advantage of learning a model—even inaccurately—is twofold: the agent would know where to go and where not to go. It knows which regions of the state space is unsafe to explore and must be avoided. It also knows which regions are unexplored and might be informative to improve the model. This brings us to another view to intrinsic reward that encourages *diversity*.

Our work is also relevant to sensitivity analysis and its use in trainig the parameters of dynamical models. After Chen et al.'s NeuralODE on training neural networks by sensitivity analysis of the network parameters, the method was successfully applied to various tasks such as learning dynamics (Rudy et al., 2019) , optimal control (Han et al., 2018), and generative models (Grathwohl et al., 2018). Our method can be seen as a mode-free sensitivity analysis in real-world systems. In NeuralODE, the gradient with respect to the parameters requries solving ODEs for both states and adjoint states that require a transition model. Since we are working directly on the physical system, we don't need to calculate the integrals forward in time. The systems itself acts as a physical ODE solver.

The importance of learning from unlabelled experiences is a known fact in animals. Many animals function efficiently soon after birth before being exposed to a massive labeled experience. Part of it might be due to unsupervised learning but the major part of the story can be a genetic heritage after years of evolution that Zador called *genomic bottleneck*. The same idea turned out to be valid in statistical learning where an automatically discovered neural network architecture peforms surpsingly well with a shared random weight (Gaier & Ha, 2019). The embedded inductive bias in the neural network architectures could be analogous to the wiring of the brain of animal babies which transfers from generation to generation by genes.

### F.1 PROOFS

Proof to the lemma on voxelization error.

*Proof.* The voxels become boxes in 3D as in fig. 15. The gradient is estimated as the distance between two points in 3D coordinates. Hence the source of voxelization error is approximating the distance between two points in 3D with the distance between the centers of the corresponding boxes to which those points belong. This error is written next to the boxes in fig. 15. The maximmum error is inversely proportional to the distance between voxels. Meaning that the voxels which are located far away will induce less voxelization error. This is intuitively clear. When two points are too distant from each other, a slight change in their position would not change the distance between them considerably. The upper bound on the error, however, occurs for a single voxel where the error is bounded by the size of the voxel. □

## G MORE RESULTS

In this section, the results of the extra experiments that were eliminated from the main text due to the space limit are presented.

The following figures show GP models trained by a set of directional derivatives collected during the shaking phase. The results are provided for the experiments of sections 4.2 and 4.3.

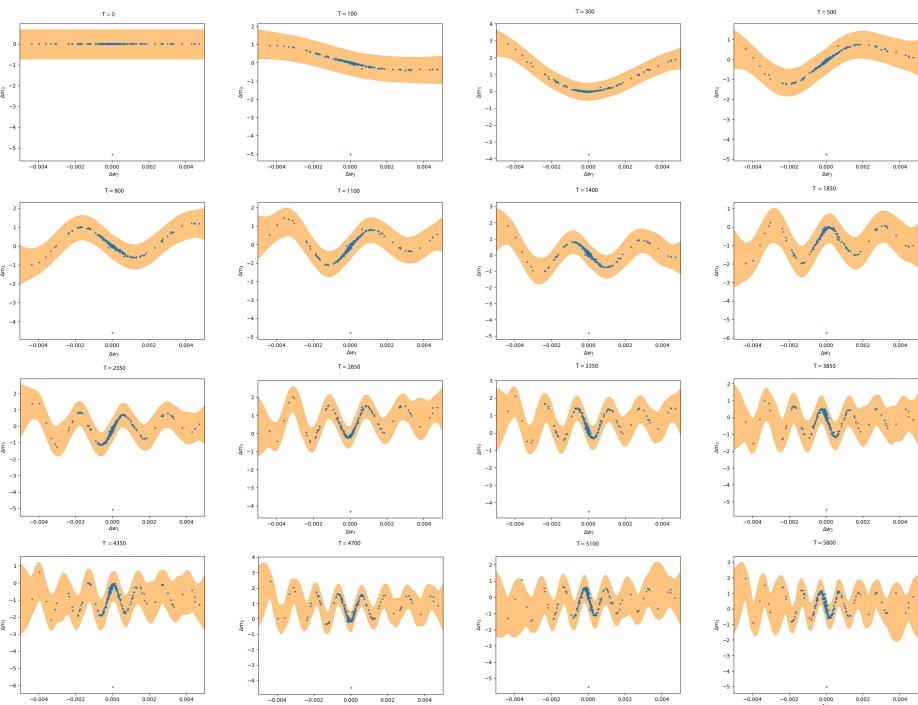

Figure 16: The time evolution of the learned GP models from directional derivatives for $\partial x_3 / \partial k_p$ by Gaussian sampling

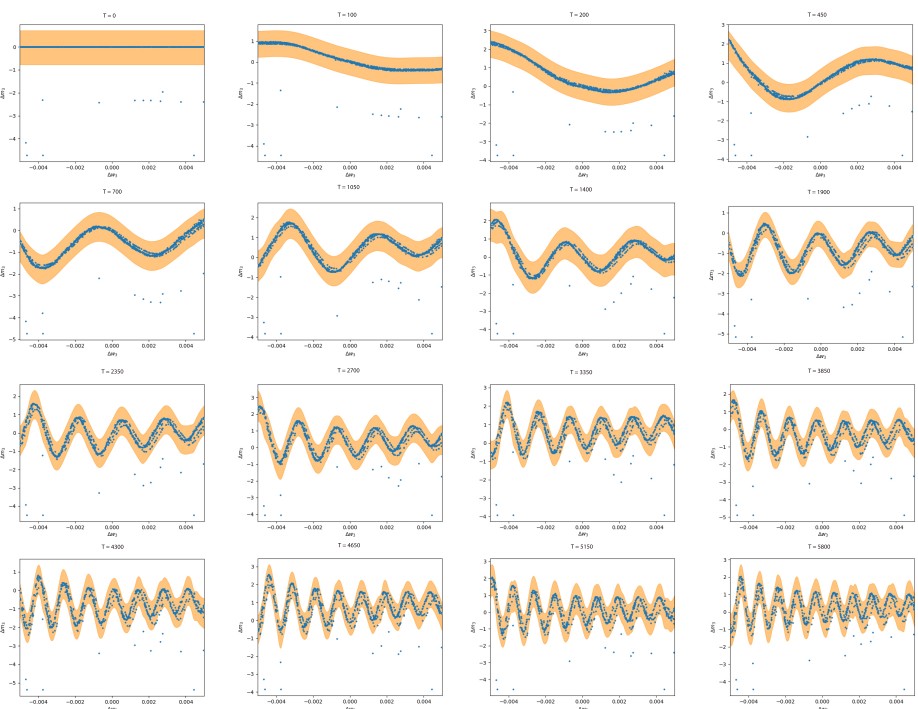

Figure 17: The time evolution of the learned GP models from directional derivatives for $\partial x_3/\partial k_p$ by uniform sampling

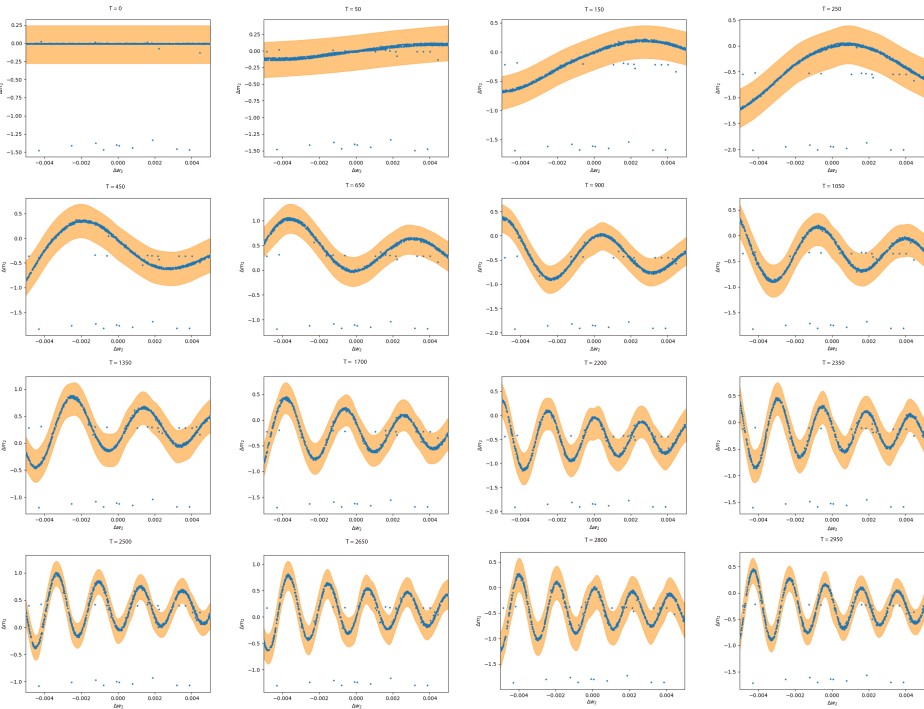

Figure 18: The time evolution of the learned GP models from directional derivatives for $\partial x_2/\partial k_p$ by uniform sampling

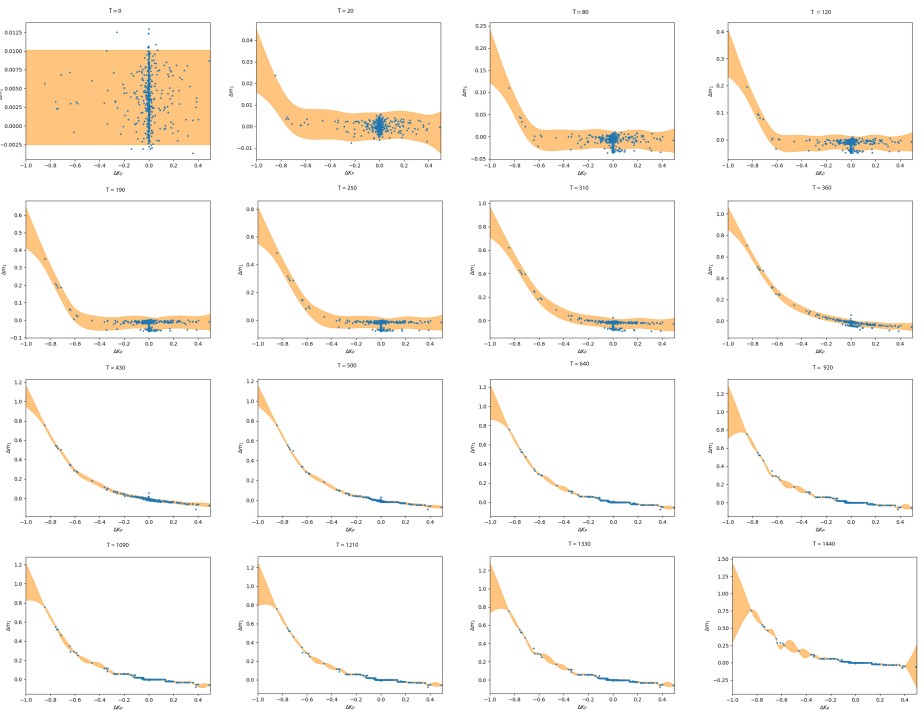

Figure 19: The time evolution of the learned GP models from directional derivatives for $\partial x_1/\partial k_p$ by Gaussian sampling

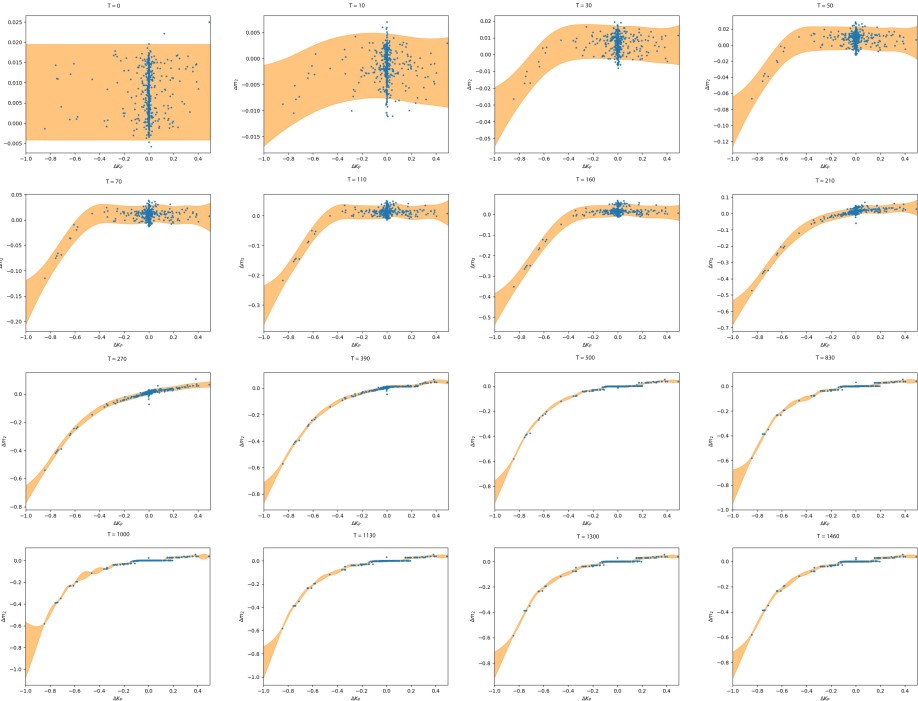

Figure 20: The time evoltuion of the learned GP models from directional derivatives for $\partial x_2/\partial k_p$ by Gaussian sampling

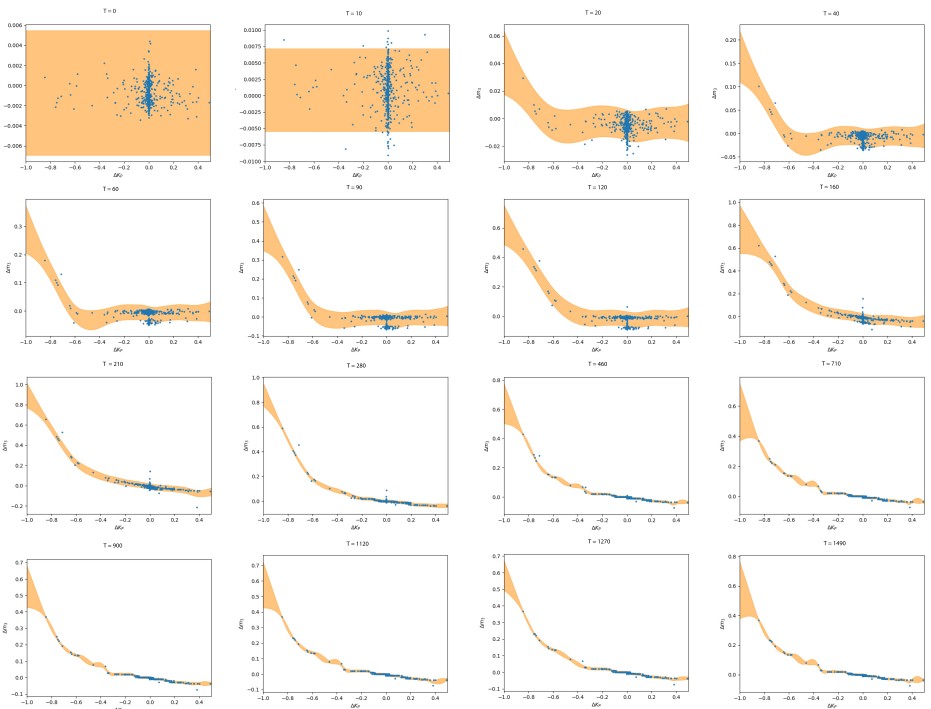

Figure 21: The time evolution of the learned GP models from directional derivatives for $\partial x_3 / \partial k_p$ by Gaussian sampling

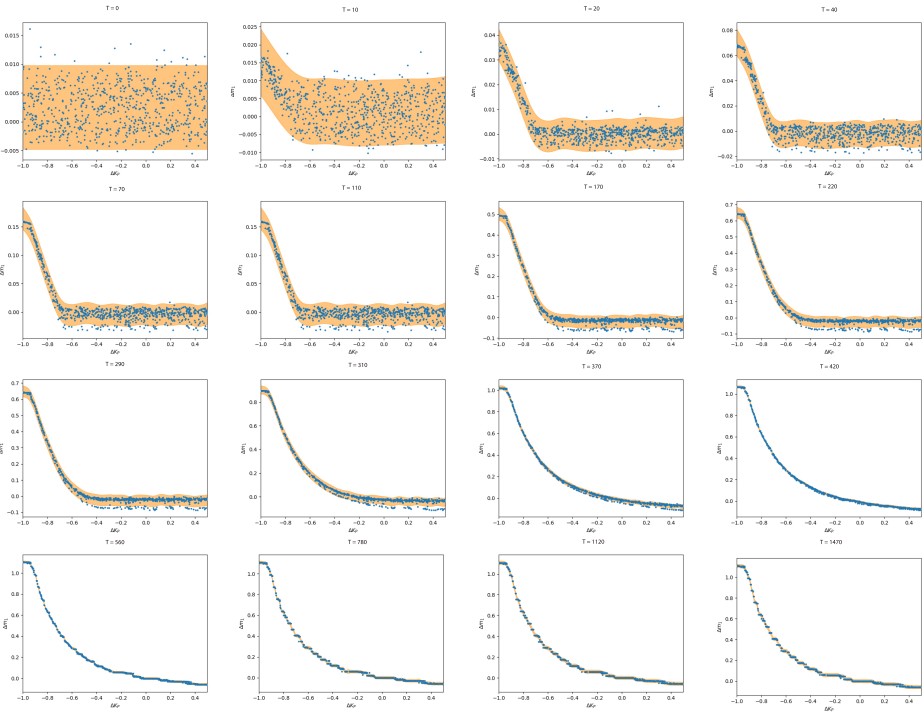

Figure 22: The time evolution of the learned GP models from directional derivatives for $\partial x_1 / \partial k_p$ by uniform sampling

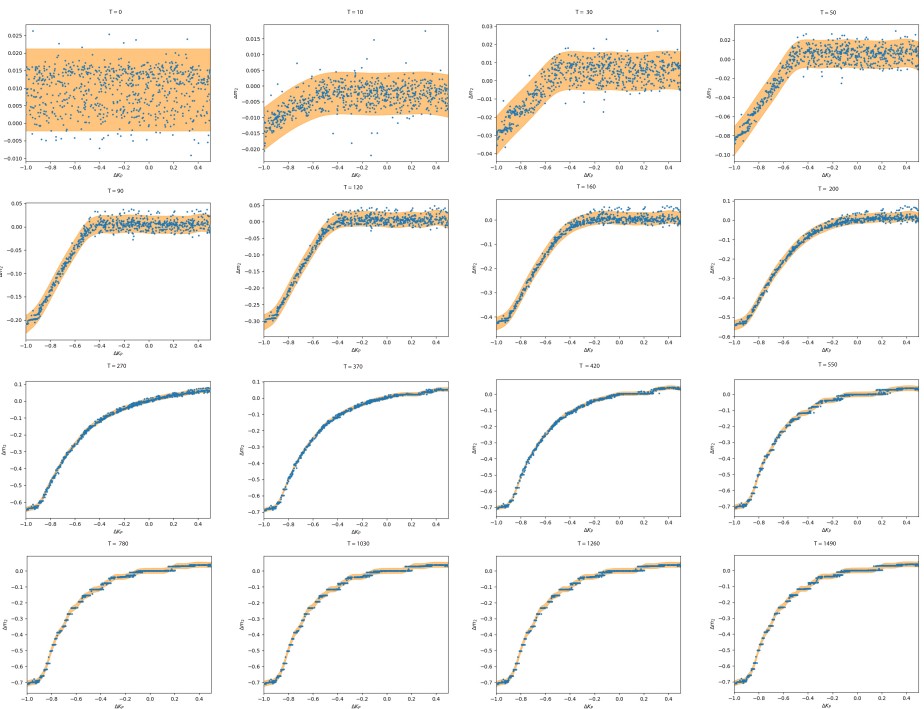

Figure 23: The time evolution of the learned GP models from directional derivatives for $\partial x_2/\partial k_p$ by uniform sampling

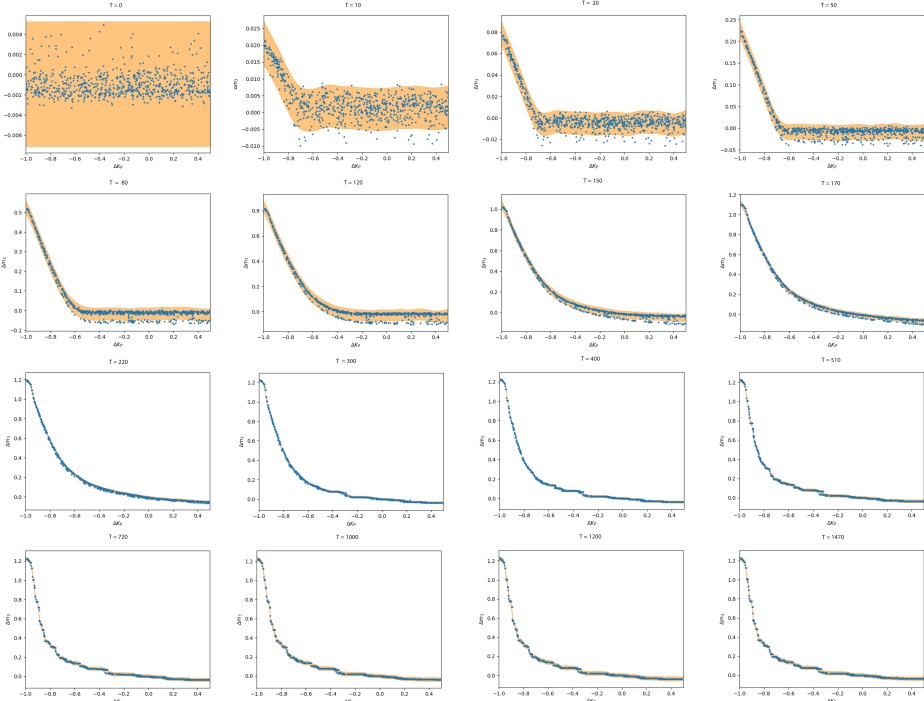

Figure 24: The time evolution of the learned GP models from directional derivatives for $\partial x_3/\partial k_p$ by uniform sampling

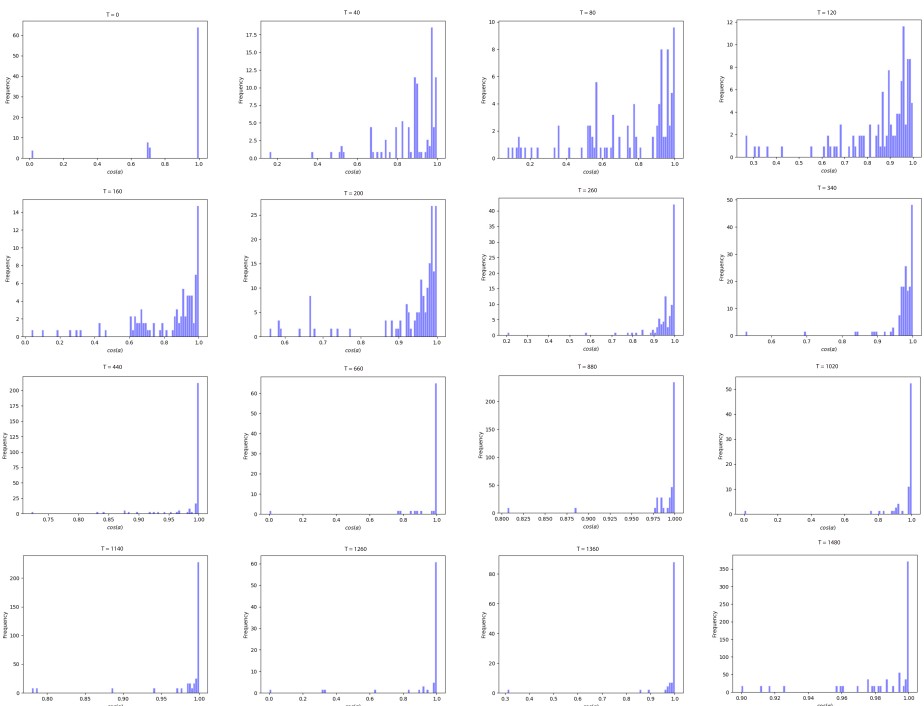

Figure 25: The time evoltuion of the histogram of $\cos(\alpha)$ where $\alpha$ is the angle between the true and predicted directional derivative. The perturbations in the training phase are generated by Gaussian sampling.

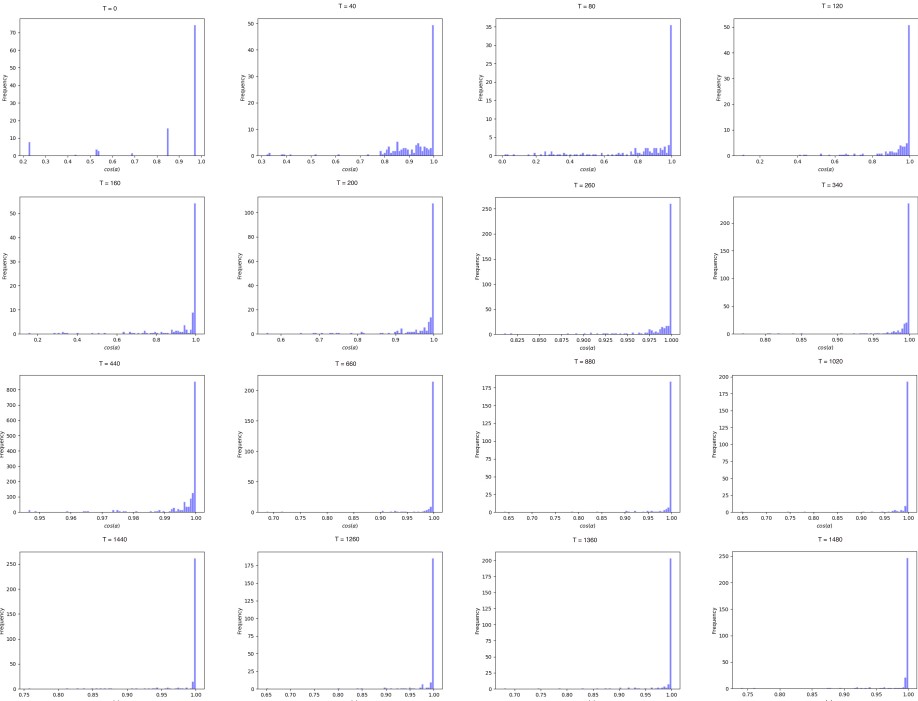

Figure 26: The time evolution of the histogram of $\cos(\alpha)$ where $\alpha$ is the angle between the true and predicted directional derivative. The perturbations in the training phase are generated by uniform sampling.

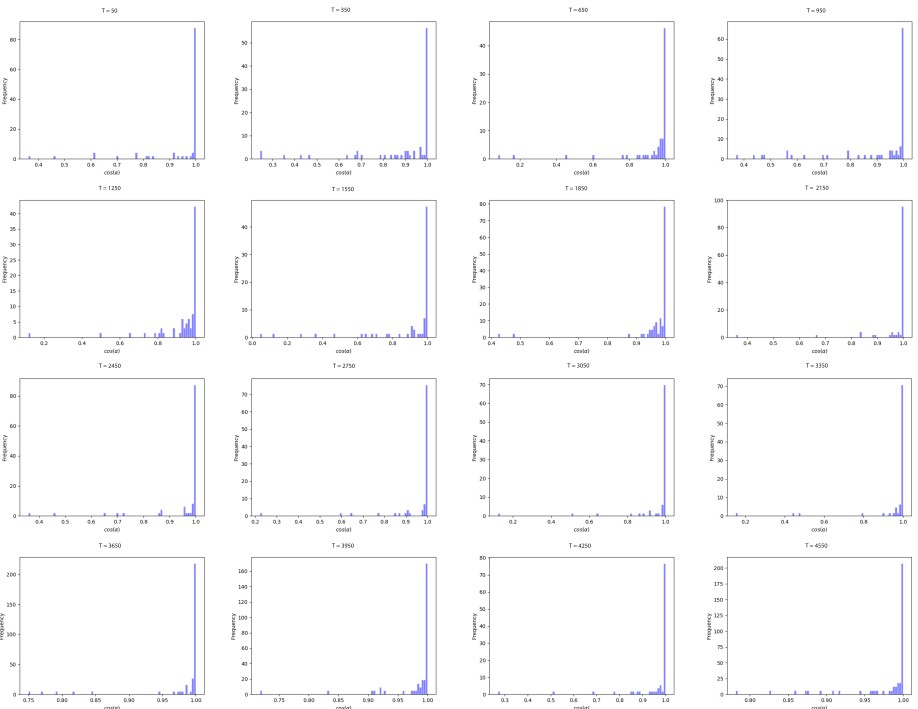

Figure 27: The time evolution of the histogram of $\cos(\alpha)$ where $\alpha$ is the angle between the true and predicted directional derivative. The perturbations in the training phase are generated by Gaussian sampling.

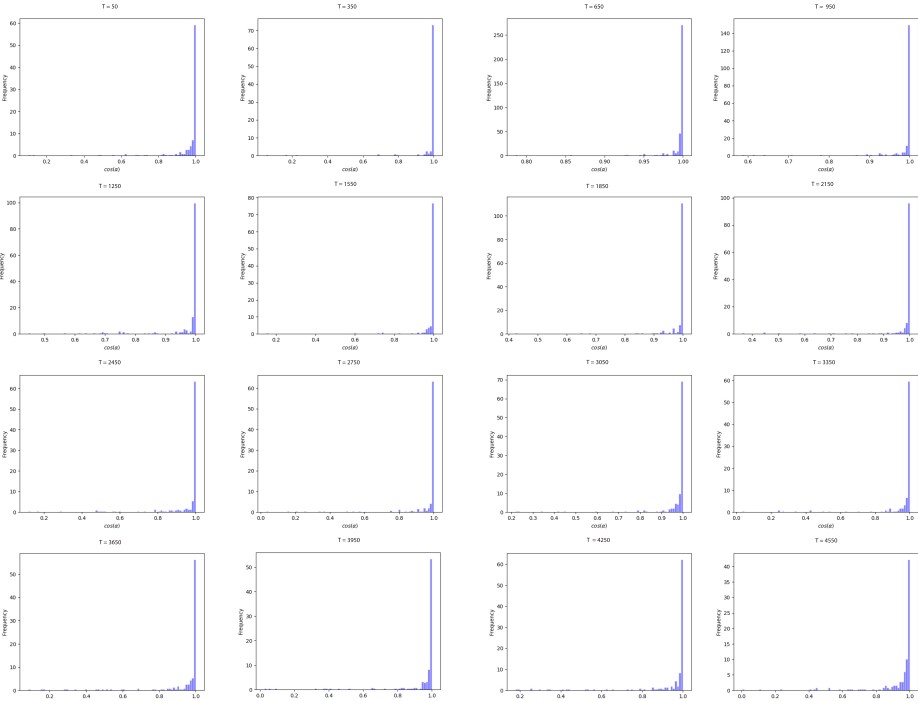

Figure 28: The time evolution of the histogram of $\cos(\alpha)$ where $\alpha$ is the angle between the true and predicted directional derivative. The perturbations in the training phase are generated by uniform sampling.

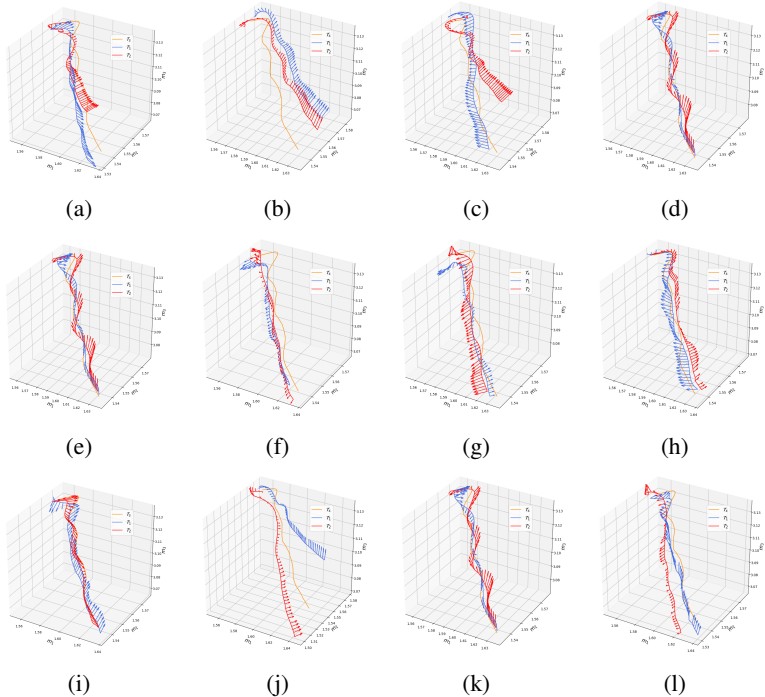

Figure 29: Examples of trajectories produced by the perturbed controller and the computed derivatives along the trajectory. The arrows are plotted as they originate from the perturbed trajectories only for easier distinction. Each arrow corresponds to the change of the states at a certain time step on the source trajectory as a result of perturbing the policy. Each figure corresponds to a pair of nominal values of $\{w, b\}$ for the linear open-loop controller of section 4.1 and the perturbed trajectories are produced by Gaussian sampling.

