# OpenReview forum: "Learning by shaking: Computing policy gradients by physical forward-propagation"
_ICLR.cc/2020/Conference — Reject_

### Official Review · AnonReviewer2 · 2019-10-23
**Official Blind Review #2**

**Rating:** 1

**Review:**

This paper addresses a very good question - can we do better in terms of model learning, so that we can find the much sought after middle ground between model free and model based RL. In particular, the authors ask, can we find a way to learn a model that is reward/task independent, so that a new task can be equally well handled. This is timely and the general thrust of the thinking, in terms of learning from perturbation around trajectories, is good but I am not sure the proposed methods are sufficiently well developed to merit publication. I am also concerned that the authors do not consider numerous issues with the setup that are fairly well understood as issues for system identification.

The main idea, as laid out in §1.1, is to observe that the parameter update depends mainly on the way a small perturbation in parameters is reflected as a variation in the optimal trajectory (by asking for the probability of a trajectory, this variation becomes the probability of a nearby trajectory). The authors then approach the approximation of this in terms of a discrete finite differences estimate. There are some extensions, such as using a local GP model instead of a local linear model and consideration of ways in which the system might not be exactly repeatable given initial states. These are all proper questions but there are many more important unanswered ones:

1. Starting with where the model setup begins, it is not clear why a complex nonlinear dynamical system, i.e., the typical multi-jointed robot taken as a dynamical system (so, not just kinematics and quasi-static movements), can be sufficiently well approximated using a discretised finite point set that is used at the start of §2 - how does one find the correct T, the correct step size, how does one change these for the local nature of the dynamics (some places might be smoother than others, in phase space), etc.? Even more importantly, are we assuming we know the proper state space ahead of time so that there is no history dependence due to unobserved variables?

2. As such, the authors are proposing to perform closed-loop system identification in a completely data-driven manner. It is well known that this is hard because in the absence of suitable excitation, not all necessary modes in the dynamics will be observed. The only controlled example considered, in §4.3, and subsequent discussion about 'zero-shot' generalisation is getting at this. However, neither at the conceptual level nor in terms of the detailed experiment do I see a good account of what allows this approach to learn all aspects of the dynamics of the system from just small perturbations around a closed loop trajectory.

3. In light of all this, I find the evaluation really weak. Some experiments I would have liked to have seen include - (i) a control experiment based on a standard multi-link arm to show how bad the issue of model mis-match is for the task being considered (I suspect, not much), (ii) experiments with local linearizations, and perhaps piecewise local linearizations, to show how much innovation is needed or is being achieved by the proposed advances, (iii) for us to be talking about 'zero shot' generalisation and the like, more sophisticated tasks beyond merely changing the reaching point (as I say before, it is not even clear that a good PID controller with a roughly plausible linearization is not sufficient to achieve similar effects, and certainly there is a plethora of more sophisticated baselines one could have drawn upon).

4. Some of the discussion comes across as a bit naive, e.g., we have a lemma 3 whose proof is simply a geometric argument about cubes without sufficient consideration of properties of dynamics. I don't doubt the result but in the way it is presented here, it seems shoddy.

Also, some smaller questions not properly explained:
a. How do you know which kernels for good for the GP in equations 9-10?
b.  Why should we expect the correlation procedure in §3.0.1 to always work without aliasing and what is the way to get at the suitable domain?




**Experience Assessment:**

I have published in this field for several years.

**Review Assessment: Checking Correctness Of Derivations And Theory:**

I carefully checked the derivations and theory.

**Review Assessment: Checking Correctness Of Experiments:**

I assessed the sensibility of the experiments.

**Review Assessment: Thoroughness In Paper Reading:**

I read the paper at least twice and used my best judgement in assessing the paper.

---

> ### Author Response · Authors · 2019-11-11
> **Response to Anonymous Reviewer 2**
>
> We thank the respected reviewer for taking time and closely reading our work. We'd like to bring up some points regarding the raised concerns in the following.
>
> Regarding the variation in the optimal trajectory:
> We don’t consider the reflected variation in the “optimal trajectory”. We consider the reflected variation in the "resulting trajectory".
>
> Regarding the analogy of our work with conventional system identification and the difficulty of closed-loop system identification:
> We are not identifying the system. We were aware of the difficulty of system identification and the fact that it requires a sufficiently rich signal in the input. However, our main argument was exactly proposing an alternative approach that learns something from the system without the need to recover true dynamical systems. We did not claim that we can recover all aspects of the dynamical system just from small perturbations.

---

### Official Review · AnonReviewer3 · 2019-10-26
**Official Blind Review #3**

**Rating:** 1

**Review:**

*Summary of paper*
This paper investigates the use of random perturbations applied to a robotic policy to learn a local gradient useful for policy optimization. The method aims to learn a policy directly on a real physical robotic system, bypassing both simulation models and model-free RL. Training pairs are gathered by perturbations of a starting policy, and the "gradient" is captured in a probabilistic model learned from the training data. The paper includes experiments on a custom 3-DOF robotic platform.

*Decision*
I vote for rejecting this paper. While the idea is interesting, the paper lacks precision in key areas and the method is not placed in context among related work. Further, it fails to communicate key ideas (particularly in the experiments) to a non-robotics reader. Without sufficient clarity and background, it is not suited to a general machine learning conference.

- Lemma 3, which attempts to justify the use of voxelization, and its proof are both imprecise and inadequate. To improve precision, please define "error causes by voxelization" in mathematical terms, e.g. ||c_i - x_i||. Also, while the statement of the lemma un-intuitively implies that larger voxels introduce smaller errors, the proof seems to say that larger errors will result for smaller gradients if larger voxels are used.
- Related work: How does this work relate to random search/evolutionary computation? How does it compare to performing those methods or a model-free RL method directly on the robot? How does it compare to learning using an inaccurate model for robot dynamics? Presumably there are numerous methods that have been tried in this area, so further context is needed.
- The evaluation is unclear, at least to a non-expert in robotics. A lack of quantitative evaluation further exacerbates this issue: nearly all experiments, even those with associated plots, are characterized qualitatively and without reference the performance of related methods.

- In addition to addressing the limitations above, I would encourage the authors to consider the use of experiments in simulation to thoroughly and quantitatively investigate the convergence/bias/variance of the gradient model w.r.t. #DoF of the robot, length of the trajectory, voxelization, # sampled trajectories, perturbation sampling method, and robot reliability/reproducibility

*Additional feedback*
- spelling errors throughout; please check thoroughly
- the captions/labels/etc. in most figures is far too small to read in a printed copy of the paper
- What is the intuition for the "empirical distribution p_e(T|\pi) = ..." on page 2? Is it counting the exact matches between the trajectory T and the M observed trajectories? (This may be more clear in the context of voxelization introduced later.)
- Figure 3: what are the units for \gamma? what is the time step?
- many of the figures are out of order w.r.t. their introduction in the text

**Experience Assessment:**

I do not know much about this area.

**Review Assessment: Checking Correctness Of Derivations And Theory:**

I assessed the sensibility of the derivations and theory.

**Review Assessment: Checking Correctness Of Experiments:**

I assessed the sensibility of the experiments.

**Review Assessment: Thoroughness In Paper Reading:**

I read the paper at least twice and used my best judgement in assessing the paper.

---

> ### Author Response · Authors · 2019-11-11
> **Response to Anonymous Reviewer 3**
>
> We thank the respected reviewer and try to answer the comments in the following.
>
> Regarding the use of simulation:
> Even though working with a simulator was much easier than building a physical robot and experimenting on it, we intentionally went for a real robot since the results of the simulator were not reliable. For example, the separation between spatial and temporal noise was not known for us prior to running experiments on the real robot. Real challenges of the problem are not visible before doing real experiments on real robots.
>
> Regarding the empirical distribution on page 2:
> It the sum of delta functions located on some trajectories in the dataset.

---

### Official Review · AnonReviewer1 · 2019-10-27
**Official Blind Review #1**

**Rating:** 3

**Review:**

This paper presents a method for control by estimating the gradient of trajectories w.r.t. the policy parameters by fitting a GP to a set of noisy trajectories executing the same controller. This is opposed to the majority of current RL methods that either learn a forward model or learn a policy. They argue that learning this gradient is a middle step between model-based and model-free RL. The method is shown to estimate gradients on simple policies (linear and nonlinear open-loop controllers, and a linear PD controller) for a free-space reaching robot, and update a controller to add a trajectory constraint to pass an intermediate state.

The paper does show that they can learn these derivatives on controllers from data, which is a cool proof of concept. The method to estimate gradients by “shaking” in a probabilistic way by fitting a GP to noisy trajectories is clever and interesting. But there are a few reasons why I believe this work is not ready for publication.

The paper only considers free-space reaching as a task, which is not a difficult problem as it does not have contacts. The policies considered are also very simple: an affine open-loop controller (U = Wt + B with 6 parameters), a simple nonlinear open-loop controller (U = Asin(wt) with 2 parameters) and a PD controller with 2 parameters. The motivation is not too convincing without showing some results on hard tasks: model-based RL methods work great in this setting, and are very likely to outperform the method proposed in the paper. The motivation for the proposed method avoids explicit model learning which is a similar motivation as model-free methods, so the paper should at least show that it works as a proof of concept in settings where model-free learning has some advantages, eg. environments with contacts. The paper should probably also compare to existing methods in those settings, although I understand that it might not outperform existing methods.

The results in section 4.4 which is the result of using the model to plan really shows that using the learned model to update the policy is probably not straightforward. The parameters of the PD controller that go from x_0 to x* are updated to pass a waypoint x*_t using the learned model. But in practice what this is basically doing is changing k_p to introduce a large, possibly inefficient deviation in the path from x_0 to x* that hits x*_t at time t. Directly planning for a path between x_0 to x*_t and then x*_t to x* would probably give a much cleaner path.

At a high level, the proposed method is likely to be difficult to apply on real problems because estimating the gradient of T w.r.t. pi is probably just much noisier than estimating the forward model directly, which is already a significant challenge. Perhaps one useful experiment is to somehow explicitly show how these two methods compare (eg. measure the variance of trajectory predictions of this method vs rolling out a learned forward model repeatedly).

Comments:

Equation 11 and 12 do not make sense/do not use standard notation. I suggest defining n (as a signal or a value, it is not clear at the moment) and defining a new output signal y_t instead of the x_t <- … notation. In particular, the way equation 11 is written seems to say the output x_t is a value-shifted version of the input x_t, NOT a time-shifted one.

The preliminaries section 1.1 does not discuss environment dynamics. This is significant because the paper seems to assume deterministic dynamics but this is never explicitly stated.

Voxelization as a solution to spatial noise is a bit surprising because discretizing the space throws away local gradient information, which seems valuable to the method. It would be good to understand the effect of this design decision better with an ablation.

Minor comments:
Page 9:
- constrain -> constraint
- Assuem -> Assume
- such controller -> such a controller

**Experience Assessment:**

I have published one or two papers in this area.

**Review Assessment: Checking Correctness Of Derivations And Theory:**

I assessed the sensibility of the derivations and theory.

**Review Assessment: Checking Correctness Of Experiments:**

I carefully checked the experiments.

**Review Assessment: Thoroughness In Paper Reading:**

I read the paper thoroughly.

---

> ### Author Response · Authors · 2019-11-11
> **Response to Anonymous Reviewer 1**
>
> We thank the respected reviewer for the technical comments and detailed reviews. Here, we try to answer the raised points.
>
> Regarding the simplicity of the task:
> Our main goal is to show that learning physical derivatives is feasible in practice despite all noises and challenges of real-world systems. We custom built the robot only to be able to safely run perturbation experiments with low cost. For the purpose of this paper, in total,  we collected around 10k trajectories which are not possible with more complex robots. Therefore, going to more complex robots and more complex controllers is definitely interesting, but was not the purpose of this paper. Here we showed the feasibility of computing the physical derivatives such that they generalize well and a proof of concept via a downstream reaching task.
>
> Regarding the experiment of section 4.4:
> The purpose of this experiment is to show the usefulness of the physical derivative in a downstream task to showcase one of its uses. There might be methods that carry out this certain experiment better and we did not claim that we do it better than alternative methods in every aspect. In this paper, we introduce the physical derivative quantity as a middle ground between model-free and model-based approaches and address its challenges in a real physical system and that experiment only serves as a complementary example to show that the learned physical derivatives generalize to unseen perturbations.
> Also, we would like to emphasize that the presence of physical derivatives allowed us to compute the desired perturbation in the parameters of the PD controller as simple as evaluating equation (21). This is a much cheaper computation than planning two paths from x_0 to x*_t and from x*_t to x*. This cheap computation is a payoff we gain for the initial extensive experiments to compute physical derivatives and learn the regressors from parameter perturbations to trajectory deviations.
>
> Regarding the difficulty of computing physical derivatives compared with a forward model:
> Yes, we agree that it is difficult to learn the physical derivatives and the main purpose of this paper and ideas that robustify the method against temporal and spatial noise is to combat these difficulties as an initial step towards this unexplored approach. We showed in the paper that using the proposed ideas, the physical derivatives can be learned from a real “physical system” and it can generalize well. The main reason that we carry out our experiments directly on the physical system rather than on a simulator was to face these challenges and propose solutions to them. We think this side of our work has fairly unnoticed in the reviews.
>
> Regarding voxelization and gradient information:
> It is an assumption in this work that the change in trajectories caused by inherent noise is less than the change caused by perturbing the parameters of the controller. Hence, by voxelization, even though some local deviations vanish, the remaining deviations are caused by perturbing the parameters of the controller. This assumption is valid in physical systems (robots) which are built with reasonable accuracy. If the system exhibits too much noise such that its controller effect is dominated by the noise, the noise must be alleviated at the hardware level.

---

> > ### Comment · AnonReviewer1 · 2019-11-15
> > **Thank you for your response**
> >
> > Thank you for your response. Ultimately I think this is an interesting direction, but perhaps the core idea (estimating the gradient of trajectories w.r.t. the policy parameters by fitting a GP to a set of noisy trajectories executing the same controller) is just innately difficult because of very high-variance gradient estimates and therefore it is a difficult way to do policy learning. This is what my intuition says, and what the results in the paper suggest, as they are on a simple problem - robot control in free space. I believe all of the reviewers mention simulators instead of physical systems because the paper would benefit from more thorough experiments on more difficult tasks, where the core hypothesis of the paper can be tested by injecting noise into the simulator. A real-world physical system may be more convincing in situations where the noise would be difficult to specify or understand, such as in a problem involving contacts.

---

### Official Review · AnonReviewer4 · 2019-10-31
**Official Blind Review #4**

**Rating:** 1

**Review:**

The paper proposes learning physical derivatives, the derivative of the trajectory distribution with respect to policy parameters. The proposed method estimates changes in  trajectories at a particular theta by using finite differences,
then fitting Gaussian Processes per timestep to generalize to new dtheta's. The paper then proposes techniques to robustify the process against noise in the system.
To deal with temporal noise, where trajectories are approximately equal up to a time shift, they simply
estimate the optimal shift and use the shifted version to estimate. To address more complicated noise, they assume sensitivity of trajectories to noise is small relative
to sensitivity to the parameters, and discretize the state space at a level such that trajectories that
differ primarily due to inherent noise look the same at the discretized level, while perturbed policy
parameters still lead to different trajectories. They then use the discretized trajectories to estimate
the finite differences.

Experiments illustrate how the learned predictions compared to actual resulting perturbations and
illustrate resulting trajectories on certain toy domains and a physical robotic finger. All experiments
are done with very low dimensional state and policy spaces (1-3 dimensions each).

Without much more extensive experimental validation, this paper should be rejected. While I am not aware
of any prior work on learning physical derivatives, the actual methods used are not novel in of themselves
beyond being applied towards learning derivatives with respect to the policy. As such, the method should be
of practical interest in order to be accepted. With limited experiments on a single very low dimensional
domain and no comparisons against any alternative methods, there is little evidence demonstrating the actual
effectiveness of the proposed method, especially on more complex domains and for downstream tasks.

Suggested Experiments:

- Stability in gradient estimation
It seems like it could require huge amounts of samples to be able to estimate gradients at parameters
where for which the system is not very stable, as states in at later timesteps can easily change in
hard to predict ways as the dynamics are propogated through time. This should be an especially big issue
if we do not already have good stable controllers close to the desired solution and needed to actually
conduct exploration in parameter space to solve a taask. I would appreciate more extensive evaluation
across multiple different (simulated) domains and assessing the effectiveness of gradient estimation
along random parameters.

- Dimensionality of policy parameters and state spaces
The current experiments only involve very small parameter spaces. It would be important to see how well
using finite differences and GP regression scales with a higher dimensional search space, which can be
demonstrated on varying dimensionalities of an LQR system for example. It would also be important to see
how gradient estimation scales with high dimensional state spaces even with a small parameter space (like
in the PD controller experiment in the paper).

- Direct Comparison against learning dynamics models
Using the same data, compare (with same metrics as in table 1) physical derivatives estimated with the
proposed approach against learning GP dynamics models and rolling out the perturbed policy with the learned
model. Without a direct comparison against learning dynamics models and understanding what situations
learning physical derivatives provides better estimates, it is unclear when or why one would prefer to learn
physical derivatives in this way compared to a model based approach.

- Quantitative results measuring costs of learned controllers
Despite the name of the paper and a description of how to compute a policy gradient via physical derivatives,
there are no experiments involving such policy gradient updates as far as I can tell. While one advantage of the
method (as well as model based approaches) is the ability to learn in unsupervised manner, it would be extremely
helpful to validate how well the physical derivatives are estimated in terms of how useful they are for a downstream
task, such as optimizing a controller for a cost function. Right now, experimental results lack any comparisons
to other methods or any other way to assess the effectiveness of estimating physical derivatives.
A comparison against regular RL policy gradient methods (or other model free algorithms) and model based RL
would give an idea as to whether the physical derivatives learned are actually useful.

Other questions and comments:
	- Table 1: is this evaluating the accuracy of the physical derivatives on the shaking data that it was
	  used for learning, or on a validation set? If on a validation set, would the validation perturbations be
	  drawn from the same distribution as the training set?
	- The zero shot planning experiment in section 4.4 seems very contrived. It does not seem like a useful task
	  to adjust the parameters of the PD controller in order to reach a state that isn't the target. The figures
	  illustrating trajectories are also not very convincing and unclear. Two points are labelled source state and
	  target state, but it is not clear which is the intermediate state it is supposed to reach. In any case, most
	  of the trajectories seem to vastly overshoot the target? final state, and it is hard to assess how close the
	  trajectories end up being to the intended states from a 2d representation of a 3d space. Quantitative results
	  would perhaps have been more useful in illustrating the effectiveness of using physical derivatives.
	- What is the purpose of figure 4? It does not appear to be referenced in the text and it is not clear what is
	  being shown.

Other notes not part of decision:
	Paper exceeds the 8 page recommended length
	Lots of small typos in the text



**Experience Assessment:**

I do not know much about this area.

**Review Assessment: Checking Correctness Of Derivations And Theory:**

I assessed the sensibility of the derivations and theory.

**Review Assessment: Checking Correctness Of Experiments:**

I carefully checked the experiments.

**Review Assessment: Thoroughness In Paper Reading:**

I read the paper thoroughly.

---

> ### Author Response · Authors · 2019-11-10
> **Response to Anonymous Reviewer 4**
>
> We thank the respected reviewer for the careful reading of our work and the constructive remarks. In the following, we separate the mentioned points and answer each in a distinct paragraph.
>
> Regarding low-dimensional experiments:
> This is true but it is due to the nature of the robotic arm and the controller. We don’t see this as a limitation of the method since we work with parametric controllers here not neural network policies.
>
> Regarding Novelty, practical interest and the need for extensive experimentation:
> This paper is an initial step towards learning a quantity from physical systems which is independent of a specific task. Compared to conventional RL where the reward function is an inseparable component, this work tries to test the feasibility of learning a useful quantity other than value functions.
> We believe learning physical derivatives is a fundamental learning problem and has not been addressed before. Therefore, showing its feasibility and methods to deal with its issues is a challenging problem by itself that we addressed in this work. Comparing with state of the art methods in RL and Control in downstream tasks is, of course, important but will be the next steps.
>
> Regarding exploration in the parameter space:
> Physical derivatives is an unsupervised quantity and we do not learn it to solve a specific task. It might not be a good idea to learn physical derivatives for a wide range of parameters only to solve a single RL task. One can think of physical derivatives as the derived proxy model of the system that can be useful in all RL and Control tasks not only one.
>
> Regarding the number of required samples, Yes, it can be true. However, the exact purpose of learning Gaussian Process regressors from perturbations in parameter to deviations in trajectories and showing the prediction accuracy on test data is to emphasize the generalizability of the method that was shown qualitatively in the figures 16-24 in the appendix and also in Table 1.
>
> Regarding high-dimensional policy: Yes, this is an important question but is beyond the scope of this paper as the first work in learning physical derivatives. Moreover, many practical controllers live in low-dimensional parameter space. For example 3 parameters in a PID controller or a few parameters in a nonlinear controller (e.g. bang-bang controller with a single on-off threshold).
>
> Regarding high-dimensional state space: We constructed separate predictors from the parameter perturbations to each dimension of the state space and showed the high accuracy of the predictions in Table 1. Please notice that we built a robot to conduct the experiments since we intentionally wanted to go beyond simulations and show the efficacy of the method on physical systems. One can build another robot with a higher dimensional state space that satisfies the safety requirements of the intended experiments in order to test the method in higher dimensions.
>
> Regarding direct comparison against methods that learn dynamics: We agree these are interesting comparisons but can be future steps. Learning physical derivatives from physical systems (not simulators) by itself has challenges and the goal of this paper is to address those challenges and finally showcasing one of its applications as in section 4.4.
>
> Regarding the usefulness of the method in downstream tasks:
> Please see 4.4 where the usefulness of physical derivatives for a downstream task.
>
> Regarding the content of Table 1:
> The prediction performance is evaluated on validation data which is left out of the initial training set.
>
> Regarding zero-shot experiment:
> We did not argue that using physical derivatives is the best way to perform the task of 4.4. The goal of this experiment is to show that the trained mapping from controller perturbations to trajectory deviations generalizes well and consequently can be used to tackle a downstream task.
>
> Regarding the purpose of Figure 4:
> Figure 4 shows the effect of perturbing the parameters of the controller on the trajectories of the physical system. The reference was mistakenly omitted from the text.

---

### Decision · Program_Chairs · 2019-12-19

**Decision:**

Reject

**Comment:**

While the reviewers generally appreciated the idea behind the method in the paper, there was considerable concern about the experimental evaluation, which did not provide a convincing demonstration that the method works in interesting and relevant problem settings, and did not compare adequately to alternative approach. As such, I believe this paper is not quite ready for publication in its current form.